# A Mineralogical Context for the Organic Matter in the Paris Meteorite Determined by A Multi-Technique Analysis

**DOI:** 10.3390/life9020044

**Published:** 2019-05-30

**Authors:** Manale Noun, Donia Baklouti, Rosario Brunetto, Ferenc Borondics, Thomas Calligaro, Zélia Dionnet, Louis Le Sergeant d’Hendecourt, Bilal Nsouli, Isabelle Ribaud, Mohamad Roumie, Serge Della-Negra

**Affiliations:** 1Institut de Physique Nucléaire d’Orsay, UMR 8608, CNRS/IN2P3, Université Paris-Sud, Université Paris-Saclay, F-91406 Orsay, France; ribaud@ipno.in2p3.fr (I.R.); dellaneg@ipno.in2p3.fr (S.D.-N.); 2Lebanese Atomic Energy Commission, NCSR, Beirut 11-8281, Lebanon; bnsouli@cnrs.edu.lb (B.N.); mroumie@cnrs.edu.lb (M.R.); 3Institut d’Astrophysique Spatiale, UMR 8617, CNRS/Université Paris-Sud, Université Paris-Saclay, bâtiment 121, Université Paris-Sud, 91405 Orsay CEDEX, France; rosario.brunetto@ias.u-psud.fr (R.B.); zelia.dionnet@u-psud.fr (Z.D.); ldh@ias.u-psud.fr (L.L.S.d.); 4Synchrotron Soleil, L’Orme des Merisiers, BP48, Saint Aubin, 91192 Gif sur Yvette CEDEX, France; ferenc.borondics@synchrotron-soleil.fr; 5Centre de Recherche et de Restauration des musées de France, UMR 171, Palais du Louvre, 75001 Paris, France; thomas.calligaro@culture.gouv.fr; 6PSL Research University, Institut de Recherche Chimie Paris, Chimie ParisTech, CNRS UMR 8247, 75005 Paris, France; 7Università degli Studi di Napoli Parthenope, Dip. di Scienze e Tecnologie, CDN IC4, I-80143 Naples, Italy; 8Université Aix-Marseille, Laboratoire de Physique des Interactions Ioniques et Moléculaires (PIIM), UMR CNRS 7345, F-13397 Marseille, France

**Keywords:** Paris chondrite, TOF-SIMS imaging, micro-Infrared reflectance spectroscopy, visible reflectance spectroscopy, micro-Raman, micro-PIXE, chemical composition, organic species, aqueous alteration

## Abstract

This study is a multi-technique investigation of the Paris carbonaceous chondrite directly applied on two selected 500 × 500 µm² areas of a millimetric fragment, without any chemical extraction. By mapping the partial hydration of the amorphous silicate phase dominating the meteorite sample matrix, infrared spectroscopy gave an interesting glimpse into the way the fluid may have circulated into the sample and partially altered it. The TOF-SIMS in-situ analysis allowed the studying and mapping of the wide diversity of chemical moieties composing the meteorite organic content. The results of the combined techniques show that at the micron scale, the organic matter was always spatially associated with the fine-grained and partially-hydrated amorphous silicates and to the presence of iron in different chemical states. These systematic associations, illustrated in previous studies of other carbonaceous chondrites, were further supported by the identification by TOF-SIMS of cyanide and/or cyanate salts that could be direct remnants of precursor ices that accreted with dust during the parent body formation, and by the detection of different metal-containing large organic ions. Finally, the results obtained emphasized the importance of studying the specific interactions taking place between organic and mineral phases in the chondrite matrix, in order to investigate their role in the evolution story of primitive organic matter in meteorite parent bodies.

## 1. Introduction

Among extraterrestrial materials available in our laboratories, carbonaceous chondrites (CCs) are considered privileged witnesses of the solar system’s early history. Their content in non-mineral carbonaceous matter, or more simply “organic” matter, is of particular interest. In these last decades, the organic material contained in CCs have been the object of varied and extended analytical studies that showed a large chemical diversity and a certain potential for prebiotic chemistry. This matter is generally divided into two families, the insoluble organic matter (IOM), and the soluble organic matter (SOM). This classification is related to the way the carbonaceous matter is extracted from the meteoritic samples before further analyses. This extraction is generally made by breaking down the rock grinded into powder, with highly concentrated HF/HCl solutions, but in some studies, the SOM is more gently extracted by dissolution either in an appropriate organic solvent, for example [1], or in warm water, for example [2]. The carbonaceous matter that is soluble in the aqueous acidic solution (or in the used solvent) forms the SOM, and the remaining insoluble material forms the IOM. Due to this physical–chemical separation, the IOM is typically composed of a cross-linked macromolecular organic matter, whereas, the SOM is composed of smaller species or moieties which are soluble in the solvent used for extraction (generally water, but organic solvents of various polarities were also used in some studies). 

The most commonly used and most fruitful techniques applied to determine the IOM chemical composition and structure are infrared (IR) and Raman spectroscopies [3,4,5,6], XANES (X-ray absorption near edge structure) spectroscopy [7,8], solid state nuclear magnetic resonance (NMR) [9], and mass spectrometry or NMR coupled to pyrolysis [7,10]. The SOM has mainly been analyzed by mass spectrometry techniques coupled to a chromatographic separation (GC-MS or LC-MS, gas or liquid chromatography coupled to mass spectrometry). These guided analyses generally target specific molecules; amino acids and nucleobases being often the most popular [11,12,13,14,15,16,17]. A few other studies analyzed the entire SOM extract without chromatographic separation, by Fourier-transform ion cyclotron resonance mass spectrometry (FTICR-MS) [1], or electrospray ionization mass spectrometry (ESI-Orbitrap) [18] to probe its complete molecular composition. Microprobe approaches such as two-step laser mass spectrometry (µL^2^MS) have been used to detect polycyclic aromatic hydrocarbons (PAHs) [19,20]. These studies and techniques significantly improved the characterization of the chemical composition and structure of the organic species contained in CCs (see for example [21] for a review on SOM, and [22] for a review on IOM). However, most of them were applied after extraction, i.e, after physical and chemical modifications of the initial meteoritic sample, and after losing the in-situ information about its location within the chondrite matrix and how it was intermingled with the mineral phases. A few recent studies started to explore these questions by using high in-situ spatial resolution imaging techniques such as transmission electron microscopy (TEM) [23], or combined TEM and electron energy loss spectroscopy (EELS) [24] or XANES spectroscopy [25] to obtain insights into the composition of the organic matter. All these studies pointed to a spatial correlation between the organic matter and the phyllosilicates present in chondrite matrices.

The Paris meteorite is a carbonaceous chondrite that was acquired by the Museum National d’Histoire Naturelle (Paris, France) and was classified in 2008 as a CM chondrite. Since then, some studies [26,27,28,29,30,31,32] mainly based on petrographic, mineralogical, and isotopic analyses, improved this classification to CM2, and a complete study was finally published by Hewins et al. [33] concluding that Paris was the least aqueously altered CM2 chondrite discovered so far and could be a CM2.9. Marrocchi et al. [31] favored a 2.7 classification, though. Hewins et al. [33] based their study on different analytical techniques including scanning electron microscopy (SEM) and TEM for mineralogy, inductively coupled plasma associated with absorption emission spectrometry (ICP-AES) or with mass spectrometry (ICP-MS) for elemental composition determination, and more specific analyses for oxygen, magnesium, and chromium isotopic measurements. All these techniques were either applied on polished sections, TEM sections, or on crushed fine-grained powder samples chemically modified before analyses (highly acidic dissolution for ICP analysis, for example). We published in a previous work [34], PIXE (particle induced X ray emission) and RBS (Rutherford back scattering) preliminary analyses of a millimetric fragment of Paris. The PIXE results gave an averaged elemental composition of the Paris chondrite consistent with the CM classification and with the elemental analyses performed by Hewins et al. [33]. RBS measurements described in [34] indicated a whole carbon content of the Paris sample studied of (7.7 ± 1.0) at.%, whereas Piani et al. [35] found a carbon content of 1.64 wt.% for their samples. Compared to the carbon content determined for other CM chondrites [36], the value found by Noun et al. [34] is high, whereas the one given by Piani et al. [35] seems to be a bit low. Besides, the organic content of Paris was more specifically investigated by Remusat et al. [30]. The authors measured its C/H ratio by NanoSIMS probe applied on a polished section and found it significantly lower than in other CM chondrites. IR transmission spectra and Raman spectra were measured on grain samples of Paris of a few tens of micron in size [37,38]. Some of the grains appeared to be rich in organic compounds and showed a CH_2_/CH_3_ ratio of 2.2 +/− 0.2 similar to the value found for some interstellar medium (ISM) objects [37]. Variations of the CH_2_/CH_3_ ratio were detected within a few microns in the same fragment, with higher CH_2_/CH_3_ values spatially correlated with hydrated minerals and lower values associated to the matrix rich in anhydrous amorphous silicates [38]. Another study of Paris organic matter was published and focused on amino-acids and the detection of some aliphatic hydrocarbons and PAHs using either water, acid water, or dichloromethane/methanol extractions with subsequent GC-MS analysis [2]. Amino acids were found by the authors in both the water extract and the acid hydrolysate extracted from the meteorite. The second was at least two times richer in amino-acids which confirmed that a large part of them were certainly not “free” molecules but fragments of larger molecules or macromolecules that released them when attacked and broken during the acid hydrolysis treatment. The authors also report the detection of PAHs with 3 to 5 rings and n-alkanes ranging from 16 to 25 carbons. Finally, a recent study by Vinogradoff et al. [39] analyzed FIB sections of different regions of Paris matrix by TEM and synchrotron-based scanning transmission X-ray microscopy (STXM) and XANES. Similarly to previous studies of other CCs [23,25], the authors found two morphological types of organic matter in Paris matrix. The first one corresponded to submicrometric individual organic particles similar in shape to inclusions, and the second one corresponded to a more diffuse fine-scale matter. Both were embedded in silicates and often associated to nanosulfides. The XANES analyses also indicated that the diffuse organic matter was generally richer in –COOR moieties than the submicrometric particles.

The first objective of this study was to analyze the mineral and organic composition of a raw chunk of the Paris chondrite without any preliminary preparation or extraction of the sample, and by minimizing any chemical modification induced by the analyses themselves. The aim was to obtain simultaneous chemical characterization of organic and mineral phases while preserving their spatial distribution within the meteorite fragment. The second objective was to obtain laboratory spectroscopic measurements which can be compared to data obtained by space missions as well as by remote observations of cometary dust and asteroid surfaces. This study was then carried out by coupling typical remote sensing tools (IR and visible reflectance spectroscopies) to TOF-SIMS (Time-Of-Flight Secondary Ion Mass Spectrometry), a high spatial resolution technique that was embarked on the Rosetta space mission (the TOF-SIMS instrument COSIMA), and that is planned to be performed on future collected asteroidal samples [40,41]. Both mid-IR spectroscopy and TOF-SIMS are performed in imaging mode with a spatial resolution of a few micrometers for the first and 1 to 2 µm for the latter. We also add micro-Raman and micro-PIXE measurements of the same sample area as an independent confirmation, verification, or clarification of the mineral composition, in order to have a better understanding of the mineralogical context of the organic moieties and components found by TOF-SIMS and IR. 

The recent results from the Haybusa2 space mission showed that the surface of the asteroid Ryugu is barely covered by regolith and is strewn by boulders, rocks, and pebbles [42], as was also the case of large areas of the comet 67P/Churyumov-Gerasimenko visited by the Rosetta mission [43]. Nevertheless, measurements on raw chondrite chunks (rather than powdered samples) by reflectance spectroscopies are rare in the literature. The reflectance measurements presented in this study should help to fill the gap. Moreover, as it will be shown by the analyses presented here, mid-infrared spectroscopy is very efficient at probing and following the water alteration of minerals; a very important question associated to the chemistry of the meteorite carbonaceous content, as it is confirmed and highlighted in this study. 

Besides being almost non-destructive (only the uppermost molecular layer of a studied surface is affected), TOF-SIMS, with its latest technical developments, is one of the rare techniques able to give rich information on the chemical composition and structure of the meteorite organic content in-situ, i.e., within the chondrite mineral matrix and together with a characterization of the mineral contents. TOF-SIMS has previously been used for the analysis of extraterrestrial samples, such as interplanetary dust grains [44], cometary particles from the Stardust mission [45,46,47], as well as meteoritic fragments and tiny inclusions within meteorites [48,49,50,51]. However, previous TOF-SIMS studies of primitive extra-terrestrial matter essentially focused on elements’ repartition in minerals (Si^+^, Mg^+^, Ca^+^, Fe^+^, S^−^ etc.), and a few of them identified some PAHs in the organic phases [52]. Only recently, thanks to technical advances [53], a few teams started looking more deeply at the organic signatures of CCs by TOF-SIMS [54]. In this study, we will show that it is possible to go further in organic and mineral phases’ investigation, identification, and 2D and 3D localization with TOF-SIMS imaging.

## 2. Materials and Methods

A millimetric chip without fusion crust of the Paris meteorite was provided by the Museum National d’Histoire Naturelle, Paris, France (MNHN). Two 500 × 500 μm^2^ areas were analyzed. Figure 1 shows images of the Paris millimetric fragment and the two analyzed areas. Area I included matrix and chondrules and was chosen for its mineralogical and chemical diversity. Area II was more dominated by matrix and was mainly analyzed to focus on the organic content. Area I was simultaneously analyzed by visible and IR reflectance spectroscopies, micro-Raman, TOF-SIMS with imaging, and micro-PIXE. Area II was only explored by TOF-SIMS. The depths of analysis of these different methods are: from a few hundreds of nm to a few µm for the IR and Raman spectroscopies depending on materials absorptivity, from 10 to 20 µm for micro-PIXE, the first 10 nm from the surface for static SIMS, and up to a few 100 nm for dynamic SIMS (3D imaging using argon cluster gun). The analyses were applied in the following chronological order: static TOF-SIMS, visible and near-IR spectroscopy, mid-IR spectroscopy with the synchrotron source, Raman spectroscopy, dynamic TOF-SIMS, micro-PIXE, mid-IR imaging with the Globar source, and finally, the surface roughness measurement.

### 2.1. Surface Roughness Measurements

The aim of this study is to characterize the selected areas on the raw sample, with minimal chemical and/or physical modification during its preparation. Nevertheless, the surface roughness is an important parameter to consider when dealing with TOF-SIMS analysis (see Appendix A for more details) and reflectance spectroscopy. Thus, the surface roughness of the chosen areas was measured with a 3D laser scanning microscope from KEYENCE (VK-X 200) that uses a 408 nm laser with an emission power of 0.95 mW. Chronologically, this measurement was performed after the other analytical techniques to avoid any possible alteration of the sample under the microscope laser beam. 

### 2.2. Visible and Near-IR (0.3–1.1 μm) Diffuse Reflectance Spectroscopy

The analyses were performed at IAS laboratory (Institut d’Astrophysique Spatiale, Orsay, France) using a grating spectrometer Maya2000 Pro (Ocean Optics), coupled through optical fibers to a home-made system, allowing illumination at angles higher than 35° (an angle between 45° and 60° was used) and collection at a fixed angle of 0°. The spot on the sample surface was around 1 mm in diameter, slightly larger than the 500 × 500 µm² of the analyzed area I. We refer to this first configuration as “macro-reflectance”. The same fibers, source, and spectrometer were later coupled to a visible microscope to define another configuration (“micro-reflectance”) where the spot on the sample was close to 50 µm in diameter (using a ×20 objective). This allowed the spectral analysis of individual particles or small mineralogical and compositional features on a larger surface, to be related to macroscopic remote sensing observations of solar system objects. The setup and calibrations were described in a previous study [55]. In our system the sample was placed horizontally, and the vertical position was adjusted by maximizing the diffuse reflectance signal. Reference spectra were collected using 99% Spectralon standard (Labsphere) and double-checked using a BaSO_4_ standard. The spectral resolution was fixed at 1 nm.

### 2.3. Mid-IR Reflectance Analysis

The analyses were performed at the SMIS (Spectroscopy and Microscopy in the Infrared using Synchrotron) beam-line of the synchrotron SOLEIL (France). A first series of IR measurements using the synchrotron source was performed before PIXE and SIMS measurements, using a NicPlan microscope (32× objective with numerical aperture of 0.65) coupled with a Magna 860 FTIR spectrometer (Thermo Fisher) operating in confocal reflection. A 20 × 20 µm² aperture was used to avoid diffraction effects in the minerals’ spectral absorption regions and to compensate for the low albedo of the meteorite. Spectra were recorded using an MCT (mercury cadmium telluride) single element detector in the mid-IR range (MIR: 7000–650 cm^−1^ or 1.4–15.4 µm) with a resolution of 4 cm^−1^ and with respect to a gold reference. In this first series, reflectance spectroscopy was measured on area I only, performing punctual measurements and mappings on selected regions of the area (Figure 2).

A second set of reflectance FTIR imaging measurements was performed after the PIXE and SIMS analyses, with an Agilent (model Cary 610/620) microscope available at the SMIS beam line, using the internal Globar source. FTIR imaging measurements were taken with a 15× objective (numerical aperture 0.62) placed in front of a 128 × 128 pixels FPA (Focal Plane Array) detector (MIR: 4000–850 cm^−1^ or 2.50–11.76 µm) resulting in a projected pixel size of 5.5 µm on the focal plane corresponding to a field of view of about 700 × 700 µm². All 16,384 spectra were thus collected simultaneously without moving the X–Y stage of the microscope (acquisition time of about 20 min for 1024 scans), allowing to easily map the localization of the different components in the meteorite. The spectral resolution was 4 cm^−1^. As explained by Dionnet et al. [38], the chemical and mineral components of the meteorite were detected in the IR imaging by three methods: The inspection of the main IR vibrational bands observed in the average spectra obtained in the first analysis with the synchrotron source (e.g., bands due to carbonates, amorphous silicates, and sulfates); the identification of additional bands that were present in selected spectra and whose spatial distribution deviated from those of the main components (e.g., anhydrous crystalline phases); and the study of the spatial distribution of the minima of the spectral second derivative. This last method permitted to easily remove the spectral continuum and provided a better detection of minor components (for instance, we better separated the contribution of components with overlapping bands).

### 2.4. Raman Micro-Spectroscopy

The measurements were performed at SOLEIL using a DXR Raman spectrometer from Thermo Fisher using a 532 nm laser for excitation, and delivering a power on sample lower than 0.5 mW. A 50× objective with a 25 µm pinhole was used. It produced power densities less than 200 W/mm^2^ with a laser spot below 2 µm. Similar power densities were used by Brunetto et al. [56] to avoid sample alteration. The spatial resolution of Raman spectroscopy is better than that of IR spectroscopy, due to a lower diffraction limitation. Raman spectroscopy was mainly used as an independent confirmation of the results obtained by IR, PIXE, and TOF-SIMS to reduce ambiguity of specific components identified within the analysis region. The data typically consisted of 5–10 scans of 30 seconds each. Raman spectra were recorded with a spectral resolution of 4 cm^−1^. Specific points or small maps on selected regions of interest were recorded, resulting in more than 100 spectra (see Figure 8A). 

### 2.5. Micro-PIXE

The analyses were performed at AGLAE (Accélérateur Grand Louvre d’Analyse Elementaire, France) with a 3 MeV proton micro-beam (about 10–20 μm in diameter) [57]. This beam exits a silicon nitride foil and passes into air. A flux of helium passes between the beam output window, the sample, and the detectors. In these experiments, two Si(Li) detectors, at 45° with respect to the beam direction, were used. One detector was dedicated to the detection of X-ray emission from light atoms (Na–Ca). The second Si(Li) detector was dedicated to X-ray emission from the heavy elements (Ca–Sr). In front of this detector a 50 μm aluminum filter attenuated X-rays from light elements and stopped the back scattered particles from the beam. The sample was fixed on a XYZ multi-axis stage which allowed to map large areas; the beam was perpendicular to the sample surface. Micro-PIXE measurements were performed on an area of 1 mm^2^ with a pixel size of 10–20 µm. This analysis zone surrounded area I (Figure 1). The acquisition time was around 30 minutes. The micro-PIXE analysis delivered an ion dose which approached 7.7 µC. Micro-PIXE measurements could therefore induce damages to the organic matter. Thus, this technique was applied after the first set of IR measurements and TOF-SIMS and Raman analyses were carried out on the meteorite fragment. 

With micro-PIXE analysis, it was possible to obtain an averaged elemental composition for the whole studied area [34] and a simultaneous mapping of all the elements heavier than F present in the analyzed area above their limit of detection. Moreover, PIXE measurements were directly translated into mass concentrations in µg/g (GUPIXWIN software [58,59]), and the elements mappings gave a direct access to a quantitative spatial distribution of all the detected elements. The ratios of their concentrations also gave an idea of the mineralogical composition and its location. 

### 2.6. TOF-SIMS (Time-of-FlightSecondary Ions Mass Spectrometry)

TOF-SIMS experiments were carried out with an ION-TOF V analyzer (at LAEC, Lebanon) including two ion sources and a mass spectrometer (see Appendix A for more instrument details).

TOF-SIMS analyses were performed using 25 keV Bi_3_^+^ primary ions, with a beam current around 1 pA and a total ion dose of respectively, 5 × 10^11^ ions/cm^2^ for area I, and 1 × 10^12^ ions/cm^2^ for area II. This beam permitted to acquire ion images with a good spatial resolution of around ~1 to 2 µm. In these experiments, the beam spot size ranged from 400 nm to 1 µm and the mass spectra were acquired with a FWHM mass resolution ranging from 3000–5000, e.g., 3345 for C_2_H_5_^+^ (m/z 29.04) and 5370 for Ca_2_O_2_H^+^ (m/z 112.92). The mass resolution and the good mass calibration due to the use of very well-known ions like elements and inorganic clusters, allowed to separate the isobaric masses and to determine masses with an accuracy better than ±20 ppm [60,61]. This allowed a very good separation between organic and inorganic fragments and ions. All the spectra were acquired with the mass range 1–3000 u.

In this study, a beam of argon clusters was applied in the so-called dual beam mode where alternating beams are used; one (Bi_3_^+^) as the analysis beam for collecting mass spectra and the other (argon clusters) for sputtering and exposing underlying layers [62]. The argon cluster beam consisted of very large clusters with a small energy per atom (Ar_1300_–Ar_1800_ at 2.5 to 20 keV). This beam is very efficient to desorb molecular layers (especially organic ones) without damaging the underlying material [63]. This served two important purposes:to clean the sample’s selected areas, I and II, before starting the analysis. As the first molecular layers are often due to common organic contaminations, removing these layers prior to mass spectrometry analysis was critical for this study. One objective of the argon cluster gun was to get rid of surface contamination without destroying the endogenous organic material. Measurements on the chondrules were used to monitor and optimize the sputtering time necessary to eliminate the organic surface contaminants. When the superficial deposition of organic contaminants was eliminated by the cleaning process, we observed simultaneously an increase then a plateau in ion emission rates of elemental ions such as Mg^+^, Si^+^, and Fe^+^ in positive mode, and O^−^, S^−^ and lowly complex compounds such as SiO_2_^−^ and PO_2_^−^ in negative mode. Typically, the ions related to the contamination disappeared after a few 10 s of irradiation. As for the endogenous organic material, the evolution of some ions such as CHO^+^, C_2_H_5_
^+^, C_3_H_5_^+^, C_5_H_10_^+^, C_3_H_7_NO_2_^+^, C_4_H_11_NO_2_^+^, C_9_H_12_O_2_^+^, and C_10_H_14_O_2_^+^ was followed as a function of Ar cluster bombardment dose. These ions underwent a high emission decrease (~10 times) during the first seconds of irradiation to later stabilize into a steady state for the following analysis which lasted 900 s of sputtering. This confirms that the organic matter identified and discussed in this paper is endogenous to the meteorite. For more details, the effect of the surface cleaning by bombarding with argon clusters prior to TOF-SIMS analysis has been discussed in [64].to perform depth profiling measurements by sputtering analyzed sections. To study molecules below the first few layers (dynamic SIMS), the dual beam technique was used. The beams of argon and bismuth were alternated, and the result was a three-dimensional image composed of successive 2D slices.

Finally, the procedure for analyzing each region consisted of first, sputtering the top most molecular layer with 10 keV Ar_1420_^+^ clusters beam with a 8–9 nA intensity (ion dose of 8–9 10^14^ ions/cm^2^), then the 500 × 500 μm² in the center of the cleaned region was analyzed by a 25 keV Bi_3_^+^ beam. Both areas were analyzed by TOF-SIMS in the static mode where only the top most part of the surface was analyzed (after cleaning with the argon beam). Area I was also analyzed by 3D depth profiling. In that case, a low energy electron beam (20 eV) was necessary to compensate the surface charge effect affecting the measurement. 

Area II was measured by TOF-SIMS after the different analysis techniques were performed on area I. A higher total ion dose was applied on area II to have a higher signal to noise ratio and improve the mass signature of the organic components compared to area I measurements. The mass spectra of area II mainly served to verify and confirm the signatures and results obtained by TOF-SIMS on area I.

Contrary to micro-PIXE, it is very difficult to obtain accurate quantitative results with SIMS since ionization efficiencies vary depending on the measured ion and on its chemical environment. However, TOF-SIMS measures the entire mass range in positive and negative modes for each pixel of the image, yielding a rich data set with information on the molecular composition and localization, in contrast to the elemental information obtained by micro-PIXE. Indeed, in addition to the elemental information obtained in TOF-SIMS analysis, larger ions corresponding to characteristic fragments and even molecular ions survive the sputtering process. They can be detected in the mass spectra and mapped to identify the chemical compound or mineral and obtain its spatial distribution.

## 3. Results

### 3.1. Surface Roughness

Figure 3A shows a 3D view of area I and its surroundings. The depth of the chondrules and the flatness and roughness obtained for this area were similar for area II. Figure 3B,C show a cross section of the 3D image at the level of the main chondrule and the profile of this section. The matrix presented a very low roughness and the surface of the chondrule was uneven; its depth reaching 20–25 μm. This explains why the surface of the chondrules was difficult to analyze by TOF-SIMS as explained in Appendix A. Figure 3D,E show a cross section of a large matrix region in the center of area II and the profile of this section. There was no significant difference in height for the matrix, only an inclination of 0.65° which corresponded to a height difference of 8 μm over 710 μm. All the matrix regions showed weak inclinations (angle lower than one degree) over several hundreds of microns which did not induce shadowing effects for the ionic emission (see Appendix A). The height differences in the matrix regions were lower than 10 µm over 700 µm. In the matrix regions, the surface was slightly rough. The roughness of the section shown in panels D and E was comparable to that of the matrix surface of the section shown in panels B and C. The value of this roughness was around a micrometer with a maximum of 2 µm. This value permitted to easily analyze the surface by cluster SIMS, at the micrometer level. It is low enough to not induce biases in TOF-SIMS analyses and results’ interpretations particularly.

### 3.2. Visible-NIR Reflectance Spectra

In Figure 4B, we plot the macro-reflectance VIS-NIR spectrum of the whole area I showed in Figure 4A. The spectrum has a relatively low albedo and a flat spectral slope above 0.6 µm. These characteristics are in agreement with the general properties observed on CM meteorites in this spectral range [65]. In particular, the reflectivity of ~6.5% we measure at 0.55 µm was compatible with the range observed in the literature on CMs (3–14%). We did not observe the 0.7, 0.9, and 1.1 µm bands although these are often observed in CM meteorites. These bands are due to the presence of Fe^2+^ and Fe^3+^ in specific sites of the crystalline structure of phyllosilicates and their intensity increases as a function of increasing phyllosilicate content. The absence of these bands can be either due to a low phyllosilicate content or to a scattering effect due to the fact that our surface is equivalent to a slab and reflectance bands are known to be deeper for powdered samples (volume scattering).

From the astrophysical point of view, the Paris meteorite spectrum is compatible with the reflectance spectra of C-complex asteroids [65], both in terms of color and albedo; the average albedo for C-complex is (6 ± 1)%, [66]. The visible slope is notably similar to that of Cg or Cgh asteroids. This confirms that such meteorite can be linked to C-type objects, an asteroid class considered to be among the most primitive and recently found to be the most abundant in the main asteroid belt [66]. 

The micro-reflectance spectra of ~50 µm-sized selected spots are reported in Figure 4C. These spots are chosen in different places of the area I to show the dependence of the VIS-NIR spectra on the composition. As a general trend, we observe the reflectance to increase in spots associated with chondrules or large mineral areas. Interestingly, in these spots (N and H in particular) the reflectance spectra clearly show the left shoulder and left side of the typical 1-µm band associated to anhydrous silicates (olivine, pyroxene). Other spots, associated with the matrix, have darker spectra with albedos that can be as low as 3%. This is probably due to the presence of organic matter in the matrix, but the presence of other sub-wavelength absorbing sub-micron inclusions such as metallic iron or iron sulfides may also contribute to reducing the albedo and subduing the absorption bands (see e.g., the VIS-NIR spectra of the nucleus of comet 67P/C-G, [67]. This assumption is confirmed by the micro-PIXE analysis presented below.

### 3.3. Mid-IR Reflectance Spectra

IR reflectance measurements were performed on the rough surface of the Paris sample to preserve spatial information and to allow for direct comparison with the other techniques used. Despite the surface roughness that induces poor and noisy baselines, it was still possible to measure IR spectra that can be interpreted. As it can be seen in Figure 5 and Figure 6, the reflectance spectra we obtain are dominated by “reststrahlen” bands, i.e., by “first surface reflection” or “surface scattering”. This phenomenon is expected and well explained, as documented by J. W. Salisbury [68]. The spectra measured on the rough surface of our sample are similar to reflectance spectra from cleaved surfaces of minerals or from large powdered particles (>75 µm of diameter) for which some volume scattering occurs and reduces spectral contrast for “reststrahlen” peaks, but the backscattered energy remains dominated by surface scattering. Consequently, in our measured spectra the strongest vibration bands (Si-O, S-O, C-O etc. stretching modes) occurring between 6 and 15 µm have positions and shapes that are different from those measured on finely ground powders. Besides, in the 2–6 µm range, when some volume scattering occurs, it is possible to identify O–H and C–H stretching modes which appear as troughs (instead of peaks) as expected for fine powders’ reflectance spectra. The intensities are however much lower and the bands are sometimes difficult to identify in our spectra due to the irregular base-line.

Figure 5 and Figure 6 present some of the IR spectra measured on the Paris sample with the synchrotron source. In Figure 5, the IR spectra of 6 major components detected in the matrix of the meteorite are shown: sulfate and carbonate (mapped in Figure 7a,b), iron hydroxide (found in regions A and C in Figure 2), anhydrous and hydrated amorphous silicates (mapped in Figure 7e–g), and some crystalline silicates spread throughout the matrix. Each IR spectrum corresponds to a square area of 20 × 20 µm². Therefore, most of the matrix spectra show a varying proportion of these 6 major components. Sulfate (Figure 5a) and carbonate (Figure 5b) regions overlap and most of the corresponding spectra show both signatures. In our measurements, iron hydroxide is always found together with a hydrated silicate signature (Figure 5f). This iron hydroxide presents 3 distinguishable and specific bands at ~3.1, 11.3, and 12.7 µm (indicated by arrows in the spectrum) that fits well with a goethite attribution (O–H stretching mode and O–H deformation vibration bands) [69,70]. This attribution is confirmed by Raman measurements which also indicate some possible mixing with hematite (see Section 3.4). 

IR signatures similar to those shown in Figure 5d (and to some extent 5c) largely dominate the measurements in the matrix (Figure 7e). The spectra of the “5d-type” show systematically a prominent and broad Si–O stretching band ranging from 10 to 10.25 µm (1000–975 cm^−1^), accompanied by a lower and more or less defined band on its shoulder at 11.35 µm (~880 cm^−1^). The whole band shape, its position, the absence of other associated features in these spectra and their large distribution through the matrix with slight variations in their shape and positions, are the reason why we attributed this signature to a disordered silicate structure, and more precisely, to an amorphous silicate component, as also found by Dionnet et al. [38]. Most of the identified hydrated silicate spectra (Figure 5c) have a prominent band ranging from 10 to 10.25 µm (1000–975 cm^−1^), sometimes accompanied by a lower band on its shoulder at 11.3 µm (~880 cm^−1^), and by the typical OH absorbance band with a relatively sharp feature at ~2.75 µm (~3630 cm^−1^) due to the O–H stretching mode of the hydroxyl groups that are generally in the octahedral layer of hydrated silicates [71,72]. The position of the prominent band could indicate a saponite attribution [73], but it is more likely a partially or totally amorphous phyllosilicate or to be more accurate, a hydrated amorphous silicate. This latter hypothesis is supported by the spectra of the amorphous silicates (Figure 5d) that dominate the matrix of the analyzed meteorite area. Indeed, as detailed above the band position and shape of part of the amorphous silicates spectra found in the matrix are very similar to those of the hydrated silicates, except that the hydroxyl band absorption is absent or comparatively very weak (spectrum d_1_ in Figure 5d). The other amorphous silicates (spectrum d_2_ in Figure 5d) show a stronger 11.35 µm band, indicating a less advanced stage of amorphization (or a more advanced stage of crystallization). 

In region B in the area around the X cross marked on Figure 2, the IR spectra measured are peculiar compared to other matrix spectra (g spectra in Figure 6). They present structured bands in the Si–O stretching region that are not found elsewhere in the matrix spectra of area I. These features correspond to a mixture of silicates that are difficult to individually identify, with some exceptions. For example, in spectrum g_5_ (Figure 6), the bands at 9.4, 10.4, and 11.4 µm indicate a probable contribution of a Mg-rich pyroxene (Mg/(Mg+Fe)~75% [74]. Whereas the band at ~9.2 µm (signaled by an arrow) present with variable intensities in spectra g_1_, g_2_, g_3_, and g_4_ (Figure 6) is a clear indication of a silica polymorph contribution. Moreover, a broad but weaker band appears at ~13.9 µm (signaled by an arrow) in part of the spectra measured in that X region (spectra g_1_, g_3,_ g_4_, and g_5_ in Figure 6). This band is possibly due to a spinel. This hypothesis will be supported further with micro-PIXE (Figure 9) and TOF-SIMS (Figure 12) results by the detection in that exact area of a particular content of aluminum and magnesium (and also of chromium, but only on surface). The spinel found covers a region that does not exceed an area of 50 × 50 µm² around the X cross. The silica polymorph covers a bit larger region around the X cross and overlaps the spinel region. The other silicates are surrounding the spinel, pyroxene unambiguous signature (spectrum g_5_ in Figure 6) being mainly at the bottom of the X. 

Only one chondrule has been partly explored by the first set of IR synchrotron measurements. In the bottom part of region D in Figure 2, another silica polymorph signature is found at 9 µm (spectrum h_1_ in Figure 6). No pure crystalline mineral was measured, the spectra obtained rather correspond to amorphous silicates at different amorphization/crystallization stages and variable compositions: spectrum h_2_ in Figure 6 probably corresponds to a partially amorphized silicate with an enstatite composition, whereas spectra h_3_, h_4_, and h_5_ would be closer to an olivine composition [75,76].

Finally, in region A and C in Figure 2, some IR spectra showed a small absorption band around 3.33–3.51 µm (2850–3000 cm^−1^), generally situated in the shoulder of a more intense OH absorbance band related to a hydrated amorphous silicate contribution (right panel in Figure 23). This small absorption, not always easy to distinguish from the noisy base-line in that spectral region, is due to C–H stretching vibrations and thus attributed to an organic content in the matrix. It is more precisely co-located with part of the hydrated amorphous silicates. As explained above, the spectra are measured in the reflectance mode on the rough meteorite fragment. The C–H (and O–H) stretching vibrations observed are necessarily due to volume scattering. Even though the C–H (and O–H) band intensity is very small compared to the Si–O bands and to what is obtained in transmission measurements [37,38], the organic local content must be quite important to be observed in these reflectance spectra. 

The second set of IR measurements with the Globar source and the FPA detector applied to area I allowed to map part of the previously discussed components over the whole area I (The IR range being slightly reduced at high wavelengths, the identified iron hydroxide could not be precisely mapped with this second set of measurements. Also, the signal to noise is not good enough to allow an unambiguous mapping of the C–H stretching bands). Figure 7 shows the mappings of the carbonate (Figure 7a), the sulfate (Figure 7b), the silica of the matrix (Figure 7c), and the Mg-rich pyroxene (Mg/(Mg+Fe) > 80%, Figure 7d) components. Panel e maps the band at 10–10.25 µm that corresponds to the amorphous silicate signature (hydrated and anhydrous ones). It is followed by the mapping of the band at ~6.25 µm that correspond to an H–O–H bending of water molecules (with maybe some contribution from a C=C stretching band) and the band at ~2.75 µm that corresponds to the O–H stretching of hydroxyl groups of the hydrated amorphous silicates (with a contribution of the O–H stretching mode of water molecules trapped in the silicate structure). The H–O–H bending mode could correspond to water adsorbed on the surface, but in this case, we would expect it to be more homogeneously distributed over area I, whereas it is here correlated to the amorphous silicate Si–O band (Figure 7e). Thus, panel f is the mapping of molecular water contained in part of the amorphous silicate structure. A difference in the hydration nature probed can explain that both f and g mappings do not exactly coincide. Moreover, contrary to the H–O–H bending mode, the O–H stretching mode is affected by volume scattering. The first one (Figure 7f) rather indicates the mapping of the hydrated silicate on the upper analyzed surface, whereas the second one (Figure 7g) gives indication of the hydrated amorphous silicates situated more in depth in area I. However, for both bands, we can clearly see the overlapping with the amorphous silicate signature; O–H and H–O–H mappings showing the aqueous alteration of part of the amorphous silicate phase.

### 3.4. Micro-Raman Spectra

Figure 8A is an optical image of area I that shows the spots (yellow numbers) and regions (yellow rectangles) measured by Raman spectroscopy. In Figure 8B, we report some selected spectra corresponding to the components most commonly observed in our Raman analysis. Raman spots are about 1–2 µm in diameter, so that in some locations two or three components can be detected in the same collection spot. Using literature and database spectra (RRUFF project database [77,78]) we were able to identify forsterite and enstatite in chondrules, and calcite, calcium sulfate, iron hydroxide, and the typical D and G bands of polyaromatic carbons commonly observed in the IOM of carbonaceous chondrites (see e.g., [4,79], and references therein), in the matrix. Raman spectroscopy is particularly sensitive to polyaromatic carbon (Figure 8B, point 15) and this feature is found in the majority of the spectra recorded all over the matrix. The average G band peak position and FWHM are 1590 ± 5 cm^−1^ and 89 ± 6 cm^−1^ respectively, compatible within error bars with values previously measured by Merouane et al. [37] on two enlarged fragments (~40 and 90 μm in size) of the Paris meteorite. Besides disordered polyaromatic carbon, most of the spectra measured in the matrix, in the large region enclosing points 6, 12, 13, 14, 22, 23, 24, and 25, in the rectangles above, and on point 21, show either pure calcite (Figure 8B, point 22), or a mixture between calcite and an anhydrous calcium sulfate (Figure 8B, point 6). Some spectra measured on chondrules show either pure enstatite (Figure 8B, point 17) or pure forsterite (Figure 8B, point 10). And finally, the spectra of point 18 and its surroundings show an iron hydroxide content, more precisely a goethite with a possible hematite contribution (Figure 8B, point 18). Point 18 and its surroundings correspond to region C for IR measurements (Figure 2).

### 3.5. Micro-PIXE Results

The concentrations of different major and minor elements measured by micro-PIXE on area I and its surroundings (1 × 1 mm²) are given in Table 1. 

Figure 9 presents the mappings of Fe, Si, Mg, Ni, Ca, S, Ti, Mn, Na, K, Al, and Cr as obtained by micro-PIXE on area I of the meteorite fragment. 

If we focus on the distribution of magnesium, iron and silicon as seen by micro-PIXE on area I, clearly the chondrules of this area are poor in iron and are magnesium rich, whereas it is the opposite for the matrix surrounding them. If we map MgO/SiO_2_ and FeO/SiO_2_ ratios re-calculated with the GUPIXWIN software (Figure 10), some specific regions on some chondrules have MgO/SiO_2_ ratios around 1.34 compatible with a forsterite-like composition (orange areas in Figure 10 left panel), whereas others are compatible with an enstatite-like composition (MgO/SiO_2_ ratio around 0.67), and some specific regions in the matrix are compatible with a fayalite-like composition (FeO/SiO_2_ ratio around 2.4, green areas in the right panel of Figure 10). 

The first two images of Figure 11 show the overlay distributions of calcium and sulfur on one hand, and iron and sulfur on the other hand, in the area I of the meteorite fragment. Calcium is concentrated in two specific regions where we also find potassium and sodium (Figure 9). Sulfur is widespread in the matrix and both Ca-rich regions are also rich in sulfur. However micro-PIXE results show variable Ca/S ratios in those regions, and some spots are clearly much richer in Ca than in S. These measurements are consistent with the calcium carbonate and calcium sulfate identified by IR and Raman measurements and with the maps obtained in IR (Figure 7). 

Iron is almost everywhere in the matrix except in the left S-rich and Ca-rich region. In the bottom right, iron is particularly concentrated and coexists with sulfur in an area surrounding the second Ca-rich region. However, the regions that are the most concentrated in iron are not the ones that are the most concentrated in sulfur. They correspond to the Fe-rich spots also appearing in the right panel of Figure 10 and lacking Si. These Fe-rich spots may contain metallic iron. Indeed, a few of them also correspond to a high concentration in Ni (Figure 9), but others may also correspond to an iron hydroxide and/or oxide, possibly mixed with an iron sulfide as suggested by IR and Raman findings. 

Figure 11c,d show the overlay distributions of calcium and silicon on one hand, and silicon and sulfur on the other hand, in area I. Apart from the chondrules, two specific Si-rich spots appear. The first one, on the top left of area I is also rich in Mg (MgO/SiO_2_ ~0.8–0.9) aside from Ti, Mn, and Cr, and is just next to an aluminum-rich spot. The second Si-rich spot is in the middle left of area I. It is surrounded by a Ca-rich region and is lacking all the metals such as Mg, Fe, Cr, Mn, Ni, and Ti. Then, this Si-rich spot could correspond to a silica-rich polymorph or less likely due to its size, to silicon carbide.

Except for a particular Ni-rich spot on the right side of the principal chondrule on the right of area I, Ni is essentially widespread in the matrix and always present together with iron. Aluminum is also in the matrix with a particularly rich spot mentioned above on the top of area I, between Ca-rich regions (X cross region in Figure 2). Chromium is essentially found in chondrules, especially in the ones on the top right of area one. Manganese is also particularly concentrated in those chondrules, but it is also found in the chondrules’ surroundings and in some Fe-rich regions in the center of area I.

### 3.6. TOF-SIMS Analysis

#### 3.6.1. Elements and Minerals

From the images collected on area I, it can be seen that many elements such as Ca^+^, K^+^, Mg^+^, Si^±^, S^−^, and Fe^+^ are emitted from specific regions (Figure 12). These element mappings are mostly consistent with the micro-PIXE results. This first information helps to confirm that these elements and the corresponding minerals probed by TOF-SIMS are not due to a superficial contamination and are well representative of the first top microns of the sample. 

Silicon is, however, a bit peculiar. If we focus on this element, measurements in both positive and negative modes allow to distinguish small regions of emission which present differences of composition. Indeed, Figure 12 shows that Si^−^-rich spots are located in the center of chondrules and in the area of the particular X-cross region (Figure 2) discussed in Section 3.3, while Si^+^ and SiOH^+^-rich regions perfectly match with Mg^+^-rich ones and correspond to Mg-rich silicates. Si^−^-rich spots do not exactly fit Mg^+^ (or other metals) mapping and do not co-exist with iron; they seem to rather correspond to silica polymorphs regions, since O^-^ emission is also high in those spots. Finally, silicon oxide anions SiO_x_^−^ are only emitted from the matrix, mainly around chondrules and the X-cross region. They are partially associated to iron and aluminum and are signing the presence of another kind of silicates typical to the matrix. Indeed, SiO_x_^−^ mappings correspond perfectly to the partially hydrated amorphous silicate component mapped by IR spectroscopy (Figure 7e). Thus, TOF-SIMS measurements show that at least three different major types of Si-bearing minerals are present in area I. Silicon and silicon oxide ions have different ionization responses depending on the molecular structure and possibly on the hydration and/or amorphization state of the analyzed silicate, and this outcome could be used to distinguish different silicate phases.

The identification of calcium sulfate and calcium carbonate is another example where the analysis of the whole mass spectrum turned to be a powerful tool for chemical characterization. Indeed, the mappings (Figure 12) show that sulfur oxide anions (SO^−^, SO_2_^−^, SO_3_^−^, SO_4_^−^) are emitted from the same area as Ca^+^. Moreover, sulfur oxide ions come from the same region of the matrix giving rise to the CaSO_x_^−^ anions (mapped in Figure 12). Thus, TOF-SIMS analysis in negative mode allows the identification and precise localization of calcium sulfate in area I and confirms IR and Raman results. However, the mapping of Ca^+^ is not totally correlated to the distribution of sulfur-bearing ions, CaS^−^ and CaSO_x_^−^. The main part of CaS^−^ and CaSO_x_^−^ emission is located in the top of the image and not everywhere the maximum of calcium emission is found (see Figure 12). In fact, Raman measurements detect calcite, a carbonate also seen in IR spectra, that co-exists with the calcium sulfate in that region. For this sample, TOF-SIMS does not detect the carbonate ion CO_3_^−^. This is probably due to a specific “matrix effect” (this anion is observed when a pure crystal, a pure powder of calcite, or some other mixtures containing calcite are measured [60]). However, the absence of the carbonate ion does not mean that TOF-SIMS does not measure a carbonate fingerprint. Indeed, the positive mode spectrum corresponding to the main calcium region (at the central left of area I) is presented in Figure 14. It shows two mass distributions of calcium oxide and hydroxide cations, based respectively on Ca_2_O^+^ and Ca_2_O_2_H^+^ with a repeating pattern corresponding to CaO mass difference between peaks, in both cases. For the first distribution, Ca_x_O_(x−1)_^+^ peaks are detected up to x = 18, and for the second one, (CaO)_x_H^+^ peaks are detected up to x = 12 (Figure 13). According to a previous study [60] and measurements on standards, the first pattern is the main signature of the calcium carbonate and the second one emerges principally from the calcium sulfate. Thus, the calcium carbonate location can be deduced by comparing Ca_x_O_(x−1)_^+^ and (CaO)_x_H^+^ ion mapping to CaS^−^ and CaSO_x_^−^ ion mapping: the common location regions correspond to the sulfate, and the others to the carbonate (Figure 12). This result about the Ca-carbonate/sulfate identification and location is also supported by micro-PIXE mappings (Figure 9 and Figure 11). Additionally, TOF-SIMS and micro-PIXE elemental mappings (Figure 13 and Figure 9) show that this sulfate and carbonate region is the main “source” of potassium, but with large variations of its concentration in that region. The signals’ maxima are also different depending on the technique applied (As explained in Section 2, both techniques explore different depths of the sample). This indicates a variable amount of K in the carbonate and sulfate mixture.

The sulfur anion is not totally correlated to the sulfur oxide anions indicating that some regions of the matrix that contain sulfur are not related to the identified sulfate. These regions correspond to HS^−^-rich and Fe^+^-rich regions. They are probably associated to an iron sulfide.

Other metals such as nickel, chromium, cobalt, manganese and titanium are localized with a finer spatial resolution than with micro-PIXE measurements and with the difference that TOF-SIMS sees the uppermost molecular layers of the sample, whereas micro-PIXE correspond to a more in-depth characterization (Figure 12). For example, the biggest chondrule seen in the middle right of area I is Cr-rich as monitored by PIXE and TOF-SIMS, whereas only TOF-SIMS sees a rich Cr-spot in the X cross region (Figure 2) indicating that this chromium is particularly abundant in the top surface of that region but less abundant more in depth.

#### 3.6.2. Carbon and Organic Matter

The main carbon bearing ions observed in both areas I and II are CN^−^, CNO^−^, and C_x_H_y_^±^ ions (with x from 1 to 21 and y from 1 to 15 for both areas). The localization of these ions is presented in Figure 14 for area I, together with the mappings of other carbonaceous ions such as C_x_H_y_O_z_^±^ and C_x_H_y_S_z_O_u_^−^, and a few inorganic ions, for comparison. Overall, these carbon-bearing ions come from the matrix region where the partially hydrated amorphous silicates and iron are widespread (Figure 14). When looking into details, it can be noticed that the richest spots for each kind of carbonaceous ions have different locations. Interestingly for example, C_x_H_y_^+^ and C_x_H_y_O_z_^+^ ions’ highest emission regions are very well correlated to some of the Fe^+^ highest emission regions (Figure 14) which, based on micro-PIXE measurements, correspond to the regions the most concentrated in Fe (Figure 9 and Figure 10). According to IR and Raman analyses, these specific regions contain a mixture of iron oxide and hydroxide with possibly metallic iron and/or iron sulfide, together with amorphous silicates (Figure 7e) that are partially hydrated (Figure 7f,g) and could be Fe-rich silicates (Figure 10 and Figure 12). In negative mode however, when compared to SiO_x_^−^ mapping, it can be noticed that the richest spots containing C_x_H_y_^−^ anions appear as inclusions or nodules surrounded by a matrix of partially hydrated amorphous silicates. And finally, C_x_H_y_S_z_O_u_^−^-ions seem to be more related to the sulfate (SO_x_^−^ and CaSO_x_^−^ distributions in Figure 12) than the sulfide (HS^−^ distribution in Figure 12) components. Even though the richest spots of CaSO_x_^−^ do not completely coincide with the richest ones in C_x_H_y_S_z_O_u_^−^, at least, this co-localization seems to indicate that sulfur is oxidized in these organic moieties. 

CN^−^ and CNO^−^ ions’ localization mostly coincide with C_x_H_y_^−^ ions. However, part of CN^−^ and CNO^−^ ions are concentrated in specific small regions of a few tens of µm^2^ and less, that do not exactly match C_x_H_y_^−^ -richest spots. These small domains have also been observed in area II. When analyzed in “depth” thanks to the dual beam mode (dynamical sputtering with the Ar beam), the signal of CN^−^ and CNO^−^ remain as high as on the surface and reveals that part of these ions are coming from inclusions embedded in the matrix. An example of this distribution in the depth is shown in Figure 15 for area I. The other ions presented in this Figure, PO_2_^−^ and NH_4_^+^ are also emitted from specific volumes but only CN^−^, CNO^−^, and NH_4_^+^ have corresponding spatial distributions.

The selection of a region of interest (ROI) corresponding to the whole CN^−^ and CNO^−^-rich areas gives a surprising TOF-SIMS spectrum in positive mode, presented in Figure 16: the mass spectrum is dominated by adduct ions or fragment ions combining CN and CNO with sodium and/or potassium. Some of the major ions identified in the positive spectrum of the selected ROI are: CNNa^+^, (CNNa)Na^+^, (CNNa)_2_Na^+^, (CNONa)Na^+^, (CNONa)_2_Na^+^, (CNOK)K^+^, (CNOK)_2_K^+^, and (CNOK)CNONa^+^. This cluster signature together with the 3D analysis excludes a superficial contamination and indicates the presence of cyanide and/or cyanate salts embedded in the matrix. 

Then, if we apply a more restrictive selection by looking into the ROI that corresponds to a co-location of CN^−^ and NH_4_^+^ ions, or of CN^−^ and CH_2_N^+^ ions (both cations are located in the same C_x_H_y_^−^-rich regions (Figure 14)), the corresponding mass spectrum shows a specific signature with enhanced nitrogenated hydrocarbon and oxygenated hydrocarbon peaks (Figure 17). For example, as seen in Figure 18, some organic peaks such as CH_4_N^+^ (which can be an amino acid fragment) and C_2_H_6_N^+^ (a prominent ion in the mass spectra of many organic compounds containing nitrogen) are enhanced by a factor of three to ten, and other peaks such as CH_2_NO^+^ and CH_5_N_2_O^+^ are revealed. This restrictive selection brings out a smaller area in the CN^−^ and CNO^−^-rich inclusions described above and the enhanced peaks found in its mass spectra correspond to a nitrogen containing complex organic signature. 

To sum up, two component types give rise to CN^−^ and CNO^−^ ions. The first type corresponds to salts of hydrocyanic acid that are quite widespread throughout the matrix and surrounded by the partially hydrated silicates. It is probably indicative of traces of aqueous circulation and/or alteration. The second compound type is nitrogen-containing organic matter further discussed below.

For the ROI corresponding to C_x_H_y_^+^-rich regions, the main hydrocarbon ions are dominated by C_2n_H_2n−1_^+^ and C_2n+1_H_2n_^+^ fragments in positive mode and C_x_H^−^ and C_x_^−^ in negative mode (see Figure 14 for the distribution images of these series of ions). The same result is obtained for the ROI corresponding to C_x_H_y_^−^-rich regions. The highly unsaturated anions dominating the negative spectra also appear when PAH standards are measured in negative mode by TOF-SIMS [64], and are certainly related to the polyaromatic moieties probed by micro-Raman (Figure 8). The cations, C_2n_H_2n−1_^+^ and C_2n+1_H_2n_^+^, are much more saturated than the C_x_H^−^ and C_x_^−^ anions, however, they are less saturated than the most intense cations, C_2n_H_2(n+p)−1_^+^ and C_2n+1_H_2(n+p)+1_^+^, usually found for PAH standards measured by TOF-SIMS [64]. Indeed, we are not probing individual PAHs or a mixture of them but a much more complex structure containing polyaromatic moieties together with a huge diversity of other “complex” organic moieties containing nitrogen, oxygen, sulfur, and/or phosphorous that are also revealed in this ROI. Indeed, it is easy to detect heteroatom-containing organic ions with up to 20 atoms. Table 2 lists the most intense peaks detected in positive and negative modes and related to these species, together with their probable and plausible attribution(s) considering the measurement accuracy.

Moreover, in regions where hydrocarbon ions are located (C_x_H_y_^±^ -rich regions), even if the mass spectra are still dominated by the minerals’ signatures, they show peaks that are due to organic ions up to 650 u at least (see Figure 18 for some examples in positive mode for m/z between 150 and 420 u). As the heavier organic species are not easy to pull out intact from the matrix, and even less easy to ionize, especially in an environment dominated by mineral species, these high mass organic signatures are an important glimpse on the macromolecular and complex structure dominating the organic content of a carbonaceous chondrite. 

Some iron containing organic ions are detected in the mass spectra. The most intense ones are: CH_3_Fe^+^, C_3_H_3_Fe^+^, C_3_H_5_Fe^+^, C_4_H_4_Fe^+^, C_4_H_6_Fe^+^, C_5_H_5_Fe^+^, C_5_H_6_Fe^+^, C_6_H_6_Fe^+^, C_7_H_8_Fe^+^, C_7_H_9_Fe^+^, C_8_H_8_Fe^+^, C_8_H_9_Fe^+^, C_8_H_10_Fe^+^, and C_9_H_12_Fe^+^. As shown in Figure 19, the spatial distributions of these ions perfectly correspond to Fe^+^ and C_x_H_y_^+^ co-localizations. The detection of these ions, and especially the heaviest ones, is not only another proof of the strong spatial correlation between the organic content and iron as stated at the beginning of this section, but it may also indicate that there are strong bonds between iron and organic moieties in the meteorite matrix. 

The advantage of the ROI selection and the spatial correlation is illustrated by the next example. This time, a ROI rich in CH_2_N^+^ is selected in area II. This selection brings out a specific mass distribution above 500 u. This mass distribution presented in Figure 20 (The main peaks are detected at m/z 575.074(5), 576.077(4), 577.076(9), and 578.082(8)) is due to the ^13^C isotope and the two isotopes of copper ^63^Cu and ^65^Cu, which make it possible to assign the group of peaks to the compound [C_32_H_16_N_8_Cu]^+^ (see Appendix B for more details on the attribution). The localization of copper in area II agrees with this assumption. Indeed, the specific spot where the feature at 575.07 u is present exactly corresponds to one of the few copper-rich spots in area II (Figure 20). Figure 20 also shows that Cr and K, and to a lesser extent Ca, are the other elements particularly enriched in that specific spot, whereas the [C_32_H_16_N_8_Cu]^+^ component is only located in there. In negative mode, this specific region is also a spot very rich in CNO^−^, C_x_^−^ (7 ≤ x ≤ 12), C_x_H^−^ (7 ≤ x ≤ 12), and C_x_H_2_^−^ (7 ≤ x ≤ 9) ions. Those latter ions may indicate that the detected copper-organic ion belongs to a compound containing a polyaromatic structure, meaning that the ion could belong to a structure similar to copper phthalocyanine or its derivatives. 

It is also possible to fully identify amino-acids moieties by selecting the ROI that corresponds to a co-location of CN^−^ and CH_2_N^+^ ions. As stated above, the selected region is rich in organic matter. Thus, this choice of ROI minimizes the strong mineral contribution in the mass spectra and helps to better reveal the signature of organic compounds. To identify an amino-acid, its full fingerprint should be present in the mass spectrum. To determine this fingerprint, amino acid standards have been analyzed with the ION-TOF mass spectrometer, in the same conditions as the Paris meteorite sample. The list of these amino acids corresponds to the ones assessed in Table 3 and Table 4. The samples were obtained by a dried solution deposit on a silicon wafer. For each amino acid selected, a mass distribution fingerprint corresponding to the main fragments has been determined. These fingerprints were sought in the mass spectra to determine the presence of the molecules and not only the peak which can be attributed to the molecular ion or some fragment ion. However, we must keep in mind that the reference spectra are obtained with the pure molecule deposited on a surface. In the case of the meteorite, the molecules are embedded in a complex environment and are also, at least partly, bonded to others to form macromolecular organic entities. Their mass spectra signatures can be in that case different from the pure molecules’ ones. The various fragments and especially negative to positive ion ratios can be different. Moreover, in the standards spectra, a lot of molecular cluster ions are present and are favored by the thick sample of pure component deposited; none of these cluster ions were detected in the Paris meteorite analysis. In positive mode, almost all amino-acids show the following low mass fragments: NH_3_^+^, NH_4_^+^, C_2_H_3_^+^, CH_2_N^+^, CH_3_N^+^, C_2_H_5_^+^, CH_4_N^+^, CH_5_N^+^, CH_3_O^+^, C_3_H_3_^+^, C_2_H_2_N^+^, C_2_H_3_N^+^, C_2_H_4_N^+^, C_2_H_5_N^+^, C_2_H_6_N^+^, CH_2_NO^+^, CH_4_NO^+^, C_2_H_5_O+, C_4_H_3_^+^, C_3_H_2_N^+^, C_4_H_5_^+^, C_3_H_4_N^+^, C_3_H_6_N^+^, and C_3_H_8_N^+^. In negative mode, they almost all show the following low mass fragments: NH^−^, NH_2_^−^, CN^−^, CH_2_N^−^, NO^−^, C_2_N^−^, C_2_HN^−^, C_2_O^−^, C_2_HO^−^, CN_2_^−^, CNO^−^, CHN_2_^−^, CO_2_^−^, CH_2_NO^−^, and CHO_2_^−^. All these positive and negative fragments are detected in the ROI spectrum. The higher mass fragments are more discriminative, and a molecular attribution is sure if a maximum of fragments and the molecular peak are detected. According to this criterion, we unambiguously identify alanine, asparagine, and aspartic acid. The presence of glutamine is very likely according to the identification of a mass peak that could correspond to its molecular peak (Figure 21) and the presence of smaller fragments common to other amino acids such as C_4_H_6_NO^+^ and C_4_H_8_NO_2_^+^. Alanine and aspartic acid were also found by Martins et al. [2]. Some of the other amino-acids sought and presented in Table 3 and Table 4 are likely present but probably as fragments of larger molecules rather than “free” species, as their mass spectra behave differently from the pure free components. Therefore, these measurements would tend to indicate that these molecules are probably not “free” amino acids but rather “fragments” linked in macromolecules. 

In conclusion, without any chemical extraction, we have the signature of a complex organic matter containing a rich diversity of nonpolar and polar moieties, probably mainly macromolecular, and as stated above, located in specific partially hydrated amorphous silicate regions and intimately related to iron, and a few other metals. 

## 4. Discussion

### 4.1. Paris Elemental Composition

Figure 22 reports the elements to Si mass ratios measured in this study by micro-PIXE and normalized to CI chondrite ratios (data from [80]). The results we obtain are consistent with the averaged composition of CM chondrites obtained from Wolf and Palme [81] data, except that the Paris fragment we analyzed is particularly enriched in Ca and globally slightly richer in Mg, V, Cr, Fe, Ni, Mn, and somewhat poorer in Al. Our results are also globally consistent with Paris bulk composition obtained by Hewins et al. [33] except for Na, K, and Ca. Concerning calcium, our data are just pointing toward a particularly Ca-rich analyzed fragment, the whole region surrounding the analyzed areas being particularly rich in calcium carbonate. K seems to be underestimated by ICP-AES technique compared to ICP-SFMS (inductively coupled plasma – sector field mass spectrometry) result, however our finding is still higher than the bulk composition found by Hewins et al. [33] and is more similar to some punctual analyses made by Hewins et al. on a less altered meteorite region with laser-ablation ICP-MS. Sodium has not been analyzed by ICP-SFMS and our Na content is similar to the laser-ablation ICP-MS measurement made by Hewins et al. on a less altered meteorite region. Sulfur and chlorine were not measured by Hewins et al. in Paris bulk, however, the composition we obtain falls between the more localized measurements made by Hewins et al. with laser-ablation ICP-MS on an altered and a less altered region of Paris samples.

In conclusion, with the exception of Ca, the elemental composition of area I is quite typical of the whole Paris chondrite.

### 4.2. The Aqueous Alteration of the Matrix

The IR signatures indicate that the matrix of this Paris fragment presents a mixture of different and gradual stages of silicates’ amorphization/crystallization and hydration. This finding strengthens the results we previously published on a ~50 µm Paris matrix particle measured by IR transmission [38]. The shape and position of the prominent band of these amorphous silicates is similar to the experimental spectra obtained during an olivine amorphization [75]. Furthermore, the different amorphization/crystallization stages’ signatures going from the hydrated amorphous silicate signature to the anhydrous crystalline silicate one (forsterite is found here in the matrix in the top right of region B) are very similar to the spectra obtained by Morlok et al. [76] when applying gradual shock experiments on Murchison matrix samples. The samples studied by Morlok et al. are initially dominated by a serpentine signature which progressively turns into an amorphous and anhydrous silicate signature, and then to olivine, under very high shock pressure. 

Our IR measurements made directly on the meteorite chunk without grinding it into powder and homogenizing it, let us directly monitor this diversity in silicates structure and water and/or hydroxyl group content, and obviously show a parental/filiation link between these silicates at different hydration and crystallization or amorphization stages. Moreover, hydrated amorphous silicates and amorphous silicates (and even some crystalline silicates) are here clearly intermingled in the matrix at the micron scale. If either irradiation had transformed crystals into different states of amorphized matter, or shocks had transformed phyllosilicates into variably amorphous—hydrated or not—silicates, it should have affected the grains individually before they accreted to form the meteorite parent body. In contrast, interstitial fluids circulation in the parent body could easily explain the differential evolution and match the IR observations. Indeed, the hydrothermal alteration would affect the minerals differently depending on the local conditions encountered (porosity, chemical composition) that will impact the chemical reactivity parameters (thermal properties, pH conditions).

This conclusion is consistent with Leroux et al. [32] observations and supports their hypotheses about the aqueous alteration in Paris matrix and the way its amorphous silicates have been partially turned into hydrated ones and then eventually, into phyllosilicates. Moreover, the silicates hydration maps obtained by IR spectroscopy (Figure 7f,g) at a much larger scale (500 × 500 µm²) than TEM studies (<1 µm²), give the fluid circulation path following the amorphous silicate component throughout the matrix.

Our IR results also indicate that there are two kinds of hydration signatures in the amorphous silicates: water molecules situated in some sort of “interlayer” regions of the silicate amorphous structure, and hydroxyl groups bonded to cations (Mg^+^ and Fe^+^ mainly, according to µ-PIXE mappings). The mappings of both signatures do not exactly coincide (the brightest spots in Figure 7f do not all correspond to the brightest ones in Figure 7g and vice versa) and may indicate two different hydration stages: one would correspond to the penetration of water molecules that progressively become part of the amorphous silicate disordered structure, and the other would consist in forming the hydroxyl group bonds which implies a chemical reaction involving water molecules. The first hydration does not involve the formation of covalent bonds with water and could be favored by the porous structure reported by Leroux et al. The second is a chemical reaction that is probably triggered by acidic or basic pH conditions. However, it would be too speculative to state which process should have occurred first without further evidence from experimental simulation and kinetic studies. 

Interestingly, the hydrated amorphous silicates in the analyzed Paris fragment do not seem to be spatially related or correlated to the carbonates and sulfates found. Yet, both these components are generally associated with aqueous alteration. The carbonate found here is a calcium carbonate with the presence of some potassium. According to previous studies [82], this kind of carbonate would indicate a primary stage of aqueous alteration. At least, this would be in line with the very partial hydration of the silicates present in the matrix.

Contrary to Ca and the carbonate component, iron repartition in the analyzed Paris fragment matches well the amorphous silicates distribution. According to micro-PIXE mappings, in the matrix regions dominated by this latter, the Fe/Si atomic ratio is higher than 1 and much higher than the same ratio in chondrules (<0.2 on average). Sulfur is also present in those matrix regions with an averaged S/Si ratio of about 0.35, which implies that iron cannot be contained only in the sulfide component but must be also part of the amorphous silicate component. In matrix regions where a hydration signature of silicates is clear, the Fe/Si atomic ratio is particularly high (>2.5). That ratio could be partly explained by an enrichment in iron of hydrated silicates, as measured by Leroux et al. [32], but it is also due to the presence of the iron hydroxide/oxide mixture identified in this study. Indeed, Raman and IR measurements find goethite in specific locations in Paris matrix that could possibly be mixed to hematite at the sub-micrometric scale. Iron hydroxide and oxide are both indicative of aqueous alteration and being always associated to the hydrated amorphous silicates in the IR spectra strengthens the hypothesis of their formation during interstitial fluids circulation and interaction with the anhydrous amorphous silicate phase. Furthermore, the goethite–hematite equilibrium is a complex system that depends on temperature, water vapor partial pressure and pH conditions, among other factors. In the specific region (point 18 in Figure 8B) analyzed by Raman, the goethite signature largely dominates the possible hematite contribution which puts some constraints on the thermal and pH conditions undergone by that region of Paris meteorite. Indeed, high temperatures would have entirely transformed the goethite into hematite, and the pH conditions have probably been either acidic (pH 2–5) or basic (pH 10–14) during the aqueous alteration for the goethite to be preferentially formed [83,84,85,86].

### 4.3. Organics and Salts, Correlation with the Aqueous Alteration

We show in this study that the major source of CN^−^ and CNO^−^ ions in SIMS measurements of this carbonaceous chondrite is not strictly speaking an organic compound, but cyanide and cyanate salts; both components, organic matter and salts, being often intermingled at the micron scale. This is an important result for future interpretations of SIMS measurements on different extraterrestrial materials, and sheds a new light on some recently published results using NanoSIMS isotopic measurements on primitive meteorites and micrometeorites [87,88]. In these kinds of studies, the authors use CN^−^ peaks to determine the ^15^N/^14^N ratio and clearly reveal two different behaviors with ^15^N rich regions and ^15^N-poor ones in the same sample. If those regions correspond to different compounds, one being a cyanide or cyanate salt and the other one, organic matter, that could explain the different isotopic fractionation histories. Moreover, if the formation of the cyanate/cyanide salts is the product of water leaching of a “parent” organic matter, that would imply a complex series of chemical reactions that would involve specific functions on the organic compounds such as amides, amines, carbonates, and maybe carbamates, and specific temperature and pH conditions to occur. However, it would be more straightforward to think that these salts are directly the result of UV or ion irradiations on precursor ices composed of variable mixtures of H_2_O, CH_4_, CO, CO_2_, and NH_3_. Indeed, the infrared signature of the cyanate ion, OCN^−^, is observed in several protostellar sources (its first detection was performed in 1984 [89] and later confirmed and clarified in 1996 [90]), and laboratory experiments showed its systematic formation and persistence after ice mixtures containing NH_3_ are UV irradiated or ion bombarded at low temperatures and different energy conditions ([90,91] and references therein). Thus, the cyanide/cyanate salts found in the Paris matrix could be a direct remnant of the ices that accreted with dusts during the meteorite parent body formation. This result emphasizes the importance of organic salts’ detection and identification in primitive extraterrestrial material, as also suggested by the recent discovery of ammonium salts on the surface of comet 67P/C-G by the VIRTIS instrument onboard the Rosetta spacecraft [92]. 

According to our TOF-SIMS results, the organic matter in Paris matrix is not randomly distributed in the matrix. Organic matter is only found in regions containing iron and amorphous silicates that are partially hydrated. This result is consistent with studies of some other carbonaceous meteorites [23,25]. More specifically, TOF-SIMS analyses show that the CxHy^+^-richest regions are very well correlated to the Fe^+^ highest emission regions. According to TOF-SIMS and PIXE measurements, these specific regions most likely contain metallic iron, and/or silicates containing iron, and/or iron sulfide, and according to IR and Raman results, part of it has probably been altered by water to form goethite and possibly hematite. 

This spatial association between organic matter and some specific minerals, namely fine grained silicates, preferentially hydrated, and containing metals such as iron, reminds of the same observations on Earth, for the organic matter found in soils and marine sediments ([93], and references therein). As it has been showed for the terrestrial samples, this systematic spatial association should imply strong sorption interactions favored and allowed by the physical and chemical properties of both components.

Indeed, TOF-SIMS analyses show that the organic matter present in the Paris matrix is not only composed of carbon and hydrogen, but is rich in different heteroatoms; N, O, S and P, being the most obvious ones. The material is mainly macromolecular [22] and the ion fragments detected by TOF-SIMS are consistent with structures composed of a “nonpolar” carbon skeleton or “backbone” randomly branched and containing aliphatic and aromatic moieties; and “polar” functions (alcohols, thiols, amines, amides, nitriles, carboxylic acids, esters, ketones etc.) distributed all over the carbon skeleton. These “polar” functions or atom groups when distributed all over a macromolecule are known to form strong and long-lasting linkages between the organic component and a silicate surface via van der Waals forces. This kind of sorption is certainly widely present in Paris and other carbonaceous chondrites matrices. The binding can be even stronger if hydrogen bonds are involved, either by the presence of hydration on the silicates, or the presence of carboxylic acid or alcohol functions for example, on the organic matter. However, the most effective and strongest binding mechanisms could be:ligand exchange (for example, between –COO^−^ groups on the organic matter in acidic conditions, and the hydroxyl groups of hydrated silicates or the iron hydroxide, goethite),cation bridging (through polyvalent cations such as Fe^3+^ that form positively charged bridges between acidic groups like –COO^−^ of the organic matter and the negatively charged –OH^−^ groups on hydrated silicates edges),and ion exchange between the mineral compound and organic functions (such as primary and secondary amines, amides and alcohols, and carboxylic acids).

The spatial association of positively charged organic ions to iron, the detection by TOF-SIMS of iron-containing large organic species, and the identification of a large copper-organic compound (or fragment) indicate that those three latter binding mechanisms are also present at least to some extent in Paris matrix. This hypothesis is also supported by the recent identification of an organo-magnesium class of metalorganic compounds in the soluble organic matter extracted with methanol from meteorites (61 samples) of different petrological classes [94].

As a consequence of all these interactions involving organic functions and minerals, the organic macromolecules and molecules will adopt adequate conformations and spatial arrangements. Indeed, in small regions containing some accumulation of organic compounds, the build-up will favor a spatial arrangement that will probably present a polar outer layer and a nonpolar or less polar inner layer, forming that way, organic aggregates that could somehow resemble micelles of a few hundred nanometers in size or less. This arrangement would explain the presence and shape of the sub-micron particles of organic matter found in some carbonaceous chondrite matrices [23,25], and observed in Paris matrix [39]. Moreover, the different sorption mechanisms allowed by the partially hydrated and amorphous silicates have probably contributed to stabilize and then protect the organic components against oxidation and thermal degradation on a time scale of billions of years, in the meteorite parent body. Besides, the adsorption of organic compounds on minerals rich in metals such as iron may also have induced some chemical evolution of the organic species by facilitating (metal catalysis) chemical reactions between adsorbed moieties that could have led to increasing chain lengths and/or cross-linking rates of the organic matter. Thus, the effects of these interactions between organic and mineral compounds or phases are important to be addressed and consider, in order to better understand the whole chemical history of organic matter in primitive extraterrestrial objects and their parent bodies. 

### 4.4. Implication for Asteroids Remote Observations

Visible-infrared spectral imaging is a technique widely used in planetary science. Imaging remote sensing spectrometers have been developed in the last decades, and these instruments have been successfully used onboard different exploration missions across the solar system, e.g., De Sanctis et al. [95] for a recent example on the dwarf planet Ceres. The planetary spectral interpretation is often based on laboratory measurements of extraterrestrial materials or terrestrial analogs. The spectral study of carbonaceous meteorites can thus be regarded as a “case” for the spectral study of primitive small solar system bodies’ surfaces, as emphasized by many authors (for some recent examples see e.g., [65,73] and references therein). Our analysis provides spectral reflectance measurements of the Paris meteorite that can be useful in support of remote sensing spectral observations of planetary surfaces. These spectra will be available at the SSHADE database hosted by IPAG (Grenoble, France) [96,97]. We discuss here some implications for small bodies’ remote sensing observations.

The visible spectral range is the most studied in small solar system bodies’ spectroscopy. However, as shown in Section 3.2, the VIS-NIR range alone is a weak tool for obtaining the meteorite composition, and it is not sufficient to unambiguously discriminate among the different molecular and mineral components. Combining VIS-NIR with MIR ranges provides a much better compositional interpretation of the analyzed material. This appears to be valid both in the laboratory and in space: the combination of VIS, NIR, and MIR helps to clearly separate asteroid spectral classes and to strengthen the meteorite-asteroid links with the detection of possible biases in the extraterrestrial material collections (e.g., [98,99] and references therein).

The Paris meteorite has a VIS-NIR spectrum similar to other CMs and to Ch/Cgh-type asteroids [100] in terms of spectral shape, slope and albedo, except for the absence of the 0.7, 0.9, and 1.1 µm bands.

Thanks to the VIS and IR microscopes, we were able to detect compositional heterogeneity at a scale of about 50 µm from laboratory VIS-NIR spectra, and at 5–15 µm scale from MIR spectra. Spots with different VIS spectra correspond to zones with unambiguous signatures in TOF-SIMS, MIR, and PIXE spectra (sulfates, carbonates, silicates, etc.). It is interesting to observe how VIS-NIR spectra of some selected spots in the matrix (see Figure 4) are found to be as dark (reflectance lower than 6%) as primitive asteroids (B-types, C-types, D-types, and P-types [66], and references therein) such as Ryugu [101] and Bennu [102] currently visited by the Hayabusa2 and OSIRIS-Rex spacecrafts, and even as dark as cometary nuclei (e.g., [43,103]). In addition, chondrules’ VIS-NIR spectra resemble those of K-type and possibly S-type asteroids, both in terms of albedo (15–25%) and spectral shape (presence of the 1-µm silicate band). This reinforces the idea that primitive meteorites can be regarded as miniaturized analogs to study the heterogeneity of small solar system bodies. 

Such multi-scale spectral approach has recently been implemented also in the planetary exploration, thanks to the development of miniaturized IR spectral imaging instruments for in-situ exploration of planetary surfaces [104]. Instruments such as MicrOmega, onboard the MASCOT lander of the JAXA Hayabusa2 sample return mission, have achieved the tens of microns spatial resolution. This kind of development provides opportunity to detect small-scale compositional heterogeneity at the asteroid, similarly to what we observe on the Paris meteorite. The heterogeneous distribution of the organic materials is particularly interesting, as this can also have an impact on the potential small-scale heterogeneity at the asteroid surface sampling site. The distribution of the carbonaceous matter observed through TOF-SIMS in Paris is heterogeneous at scales <50 µm, often <20 µm (see Figure 15 and Figure 23), which implies a challenge for in-situ detection with the current asteroid missions. However, in Figure 23 we show that the detection of weak (~5% absorption in our reflectance geometry) aliphatic CHs band along with OH signatures in reflectance spectra of relatively small (~20 µm) regions, corresponds to very strong and clear organic signatures in TOF-SIMS spectra, with an abundance of carbon that can be locally (20–100 µm) as high as 30–40 at.% (the estimation is based on the whole carbon content of 7.7 ± 1.0 at.% measured for the Paris meteorite [34]). A “local” carbon atomic abundance as high then, as the average carbon content found in comet 67P’s dust [105]. This may have strong implications for remote sensing and in-situ detection of organic materials on small bodies’ surfaces. In the case of remote sensing spectral observations of asteroids, detecting a ~5% aliphatic CHs band on the whole disk (see e.g., Figure 2 in [106], for the detection of organic matter on asteroid Themis) would imply that the asteroid is extremely rich in organic materials. In the case of in-situ asteroid analysis by space missions, a ~5% detection would be a strong criterion in the search for organic-rich terrains. Different observation geometries and surface properties may also affect the band depth, though (e.g., [65]).

## 5. Conclusions

The multi-technique investigation applied in this study to analyze a millimetric fragment of Paris meteorite, has allowed to map its precise elemental and chemical compositions at a spatial scale going from 20 µm for the elements quantification to 1–2 µm for the mass spectra signatures, in-situ, without any chemical extraction or any further sample preparation, which reduces any sample modification or alteration and preserves the spatial localization information.

The Paris meteorite is the least altered CM2 classified so far, and thus, a good witness of the solar system’s early history. The matrix of Paris is found to be as dark as the very dark asteroids Ryugu and Bennu currently visited by Hayabusa2 and OSIRIS-REx [101,102]. By combining the complementary results given by micro-PIXE, TOF-SIMS and IR and Raman micro-spectroscopies, we were in particular able to show the different stages of hydration and amorphization of the amorphous silicate phase dominating the analyzed Paris fragment matrix. The infrared signatures of water and hydroxyl groups mapped for a chosen large area (500 × 500 µm²) of the meteorite fragment give a good idea of the circulation of the fluid that partially altered the amorphous silicate phase. TOF-SIMS, one of the rare techniques allowing the simultaneous and micrometric analysis and mapping of organic and mineral mixed phases, gives an interesting insight on the organic matter composition and its systematic spatial association with the partially hydrated amorphous silicate phase and iron under different states (the iron hydroxide phase found here, being of a particular interest). This latter result together with the detection by TOF-SIMS of metal-containing organic moieties emphasizes the important and specific interaction that must take place between the mineral phase and the organic material. It opens perspectives into understanding the way this interaction probably played a role in preserving the organic species from oxidation and thermal degradation, and possibly had an influence on its chemical evolution into higher molecular weights and cross-linking rates, since the Paris parent body formation.

## Figures and Tables

**Figure 1 life-09-00044-f001:**
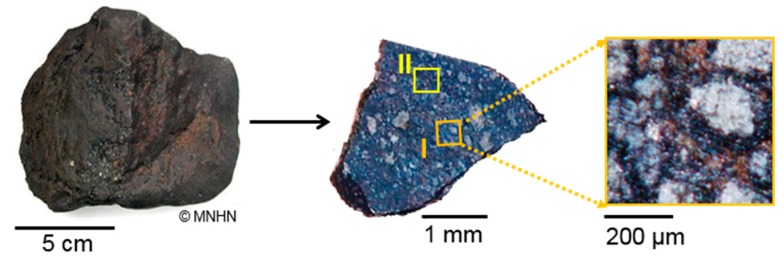
The analyzed fragment of Paris meteorite (©MNHN). The two analyzed areas are indicated with different colors and have the same size of 500 × 500 μm².

**Figure 2 life-09-00044-f002:**
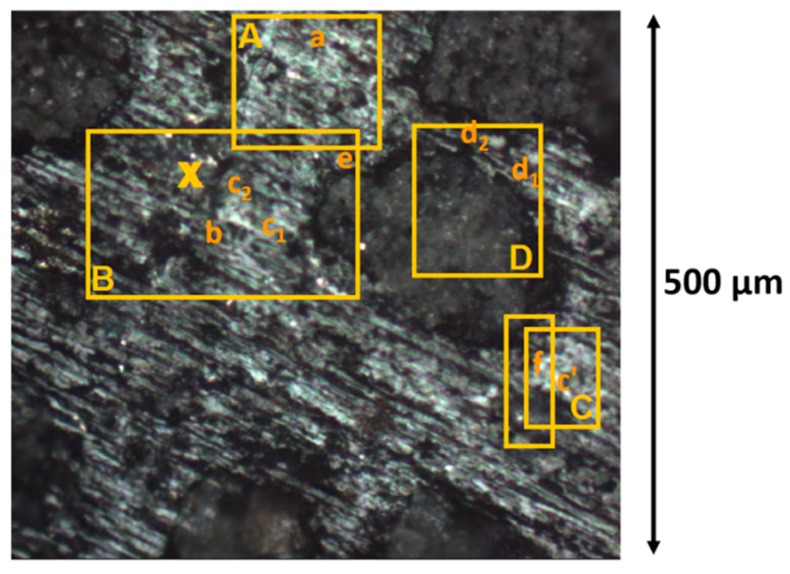
Optical image of the selected area I with in orange, the regions A, B, C, and D explored with IR reflectance spectroscopy and the synchrotron beam source. The letters a, b, c_1_, c_2_, c’, d_1_, d_2_, and f refer to the spectra shown in Figure 5 and indicate the location of each measurement. The large dark regions in the image correspond to chondrules. The “X” cross corresponds to a matrix region with a specific IR signature discussed in Section 3.3.

**Figure 3 life-09-00044-f003:**
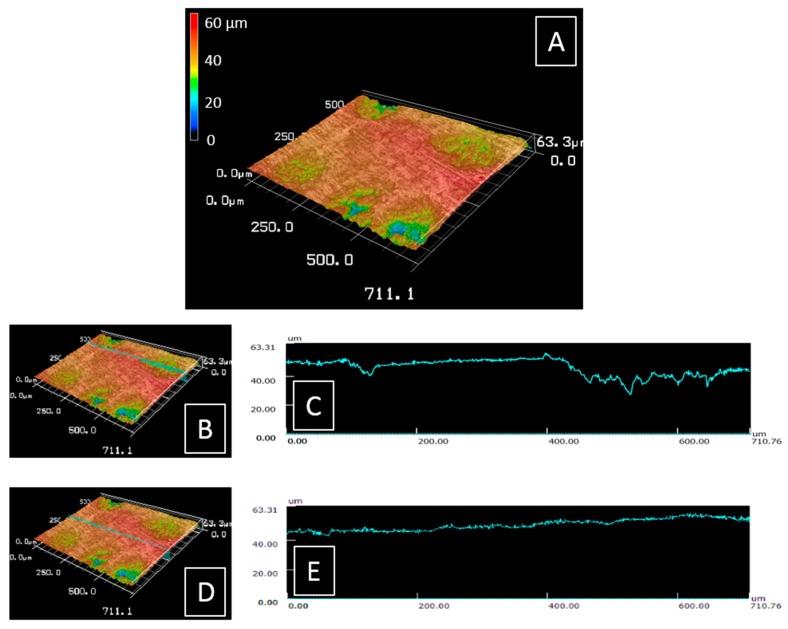
(**A**): 3D image of the surface of area I and its immediate surroundings. (**B**,**C**): A cross section passing through the large chondrule of area I, and the profile of this cross section with the matrix surface at the left and the chondrule at the right. (**D**,**E**): A cross section in the middle of the matrix, and the profile of this cross section.

**Figure 4 life-09-00044-f004:**
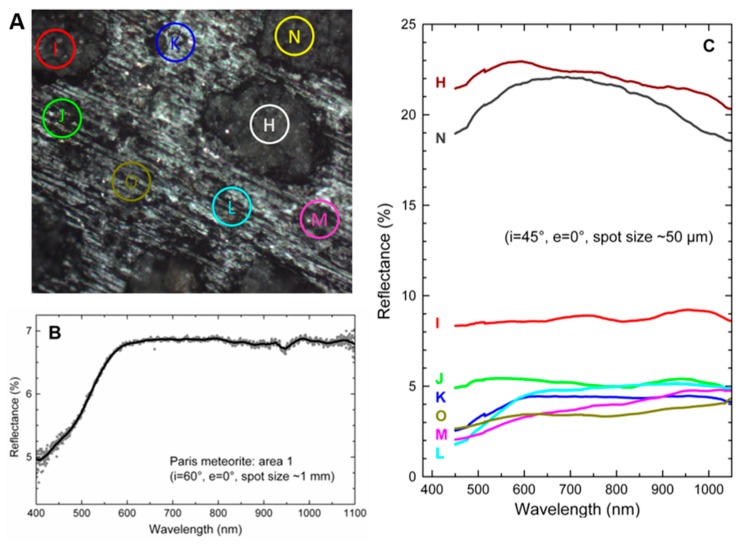
(**A**): Visible image of area I with selected spots for visible reflectance spectroscopy. (**B**): Visible reflectance spectra of the whole 500 × 500 μm² area I. (**C**): Visible micro-reflectance spectra. Each spectrum corresponds to a specific ~50 µm spot shown with the corresponding color and letter on (**A**).

**Figure 5 life-09-00044-f005:**
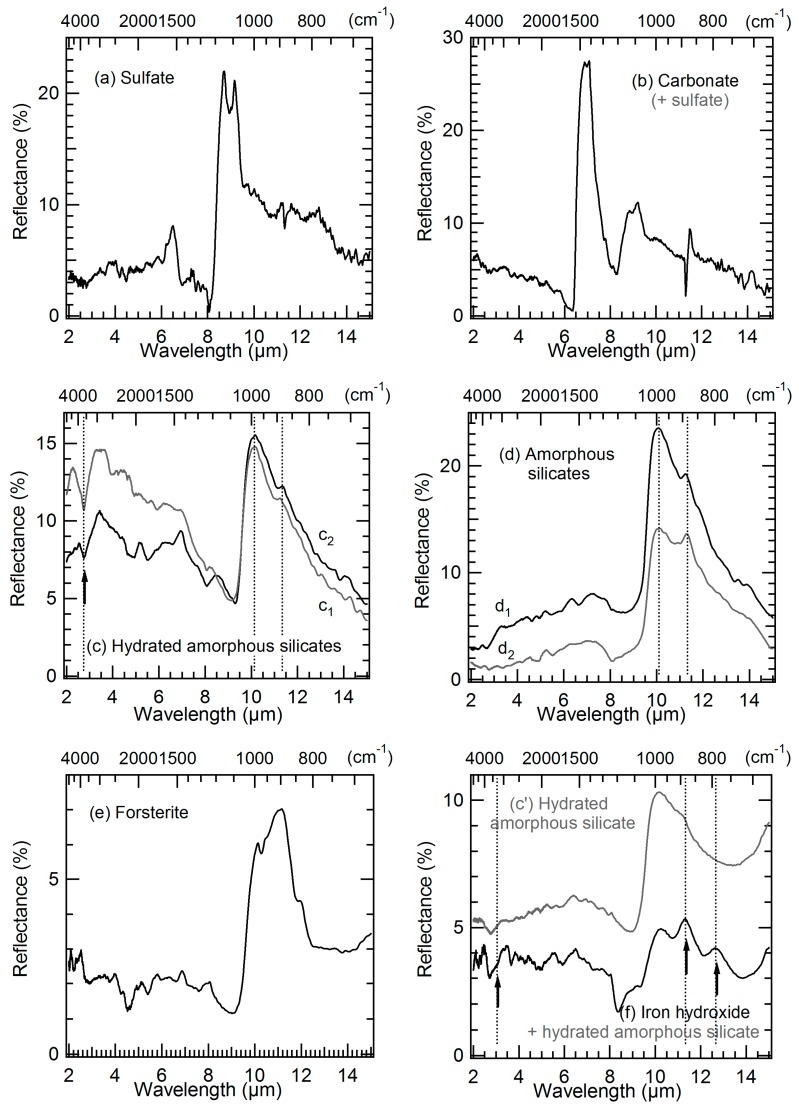
FTIR micro-reflectance spectra showing the major components found in the analyzed regions of the matrix of area I. (**a**) Sulfate spectrum. (**b**) Carbonate (mixed to some sulfate) spectrum. (**c**) Two hydrated amorphous silicate spectra (the arrow points to the hydroxyl absorption at 2.75 µm). (**d**) Two amorphous silicate spectra: d_1_ is more amorphized than d_2_ and possibly weakly hydrated. (**e**) Forsterite found in the matrix at the top right of region B (Figure 2). (**f**) An iron hydroxide (mixed to a hydrated amorphous silicate) found in regions A and C; the arrows point to its signature at ~3.1, 11.3, and 12.7 µm ((c’) is a hydrated amorphous silicate spectrum for comparison). A vertical offset has been applied on spectra d_2_ and c’ for a better visibility.

**Figure 6 life-09-00044-f006:**
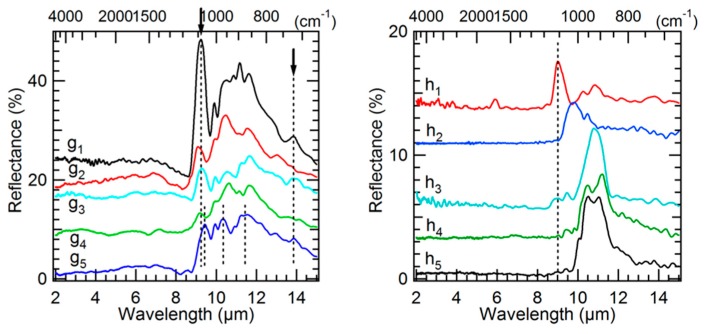
(**g**) FTIR micro-reflectance spectra of the region marked by an “X” cross in Figure 2. (**h**) FTIR micro-reflectance spectra of a chondrule region. A vertical offset has been applied on spectra g_1_, g_2_, g_3_, g_4_, h_1_, h_2_, h_3_, and h_4_ for a better visibility.

**Figure 7 life-09-00044-f007:**
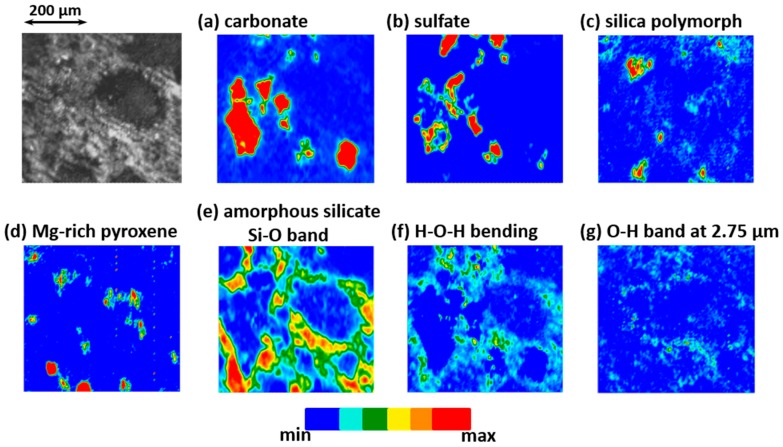
Infrared mapping of (**a**) the carbonate component, (**b**) the sulfate component, (**c**) the silica polymorph component at 9.22–9.13 µm, (**d**) the Mg-rich pyroxene (Mg/(Mg+Fe) > 80), (**e**) the amorphous silicate Si–O band at 10–10.25 µm, (**f**) the H–O–H bending band at ~6.25 µm, and (**g**) the O–H stretching band at 2.75 µm.

**Figure 8 life-09-00044-f008:**
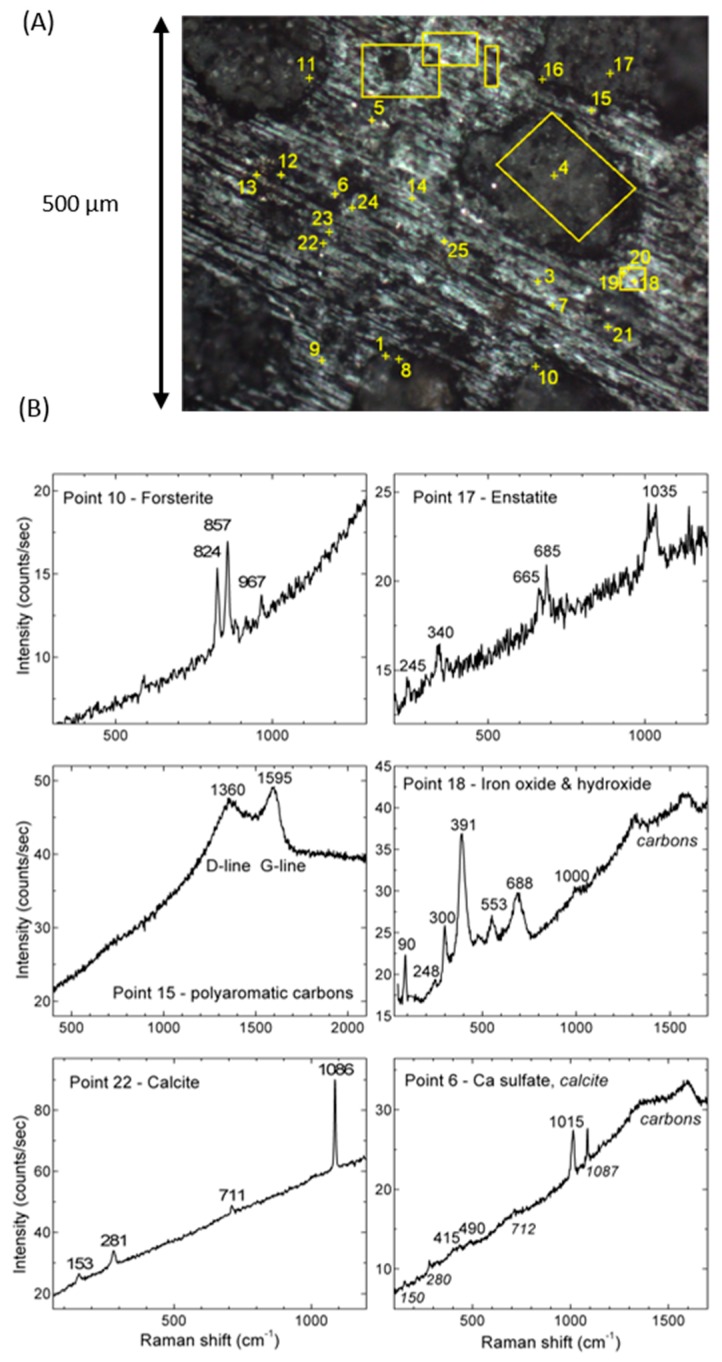
(**A**) An optical image of area I that shows in yellow the spots and regions explored by Raman spectroscopy. (**B**) Micro-Raman spectra showing the 6 components found in the analyzed points indicated by the number in panel A. (Point 15) polyaromatic carbon, (point 10) forsterite, (point 17) enstatite, (point 22) calcite (= calcium carbonate), (point 6) anhydrous calcium sulfate (mixed with some calcite and polyaromatic carbon), and (point 18) goethite (+ probably hematite contribution).

**Figure 9 life-09-00044-f009:**
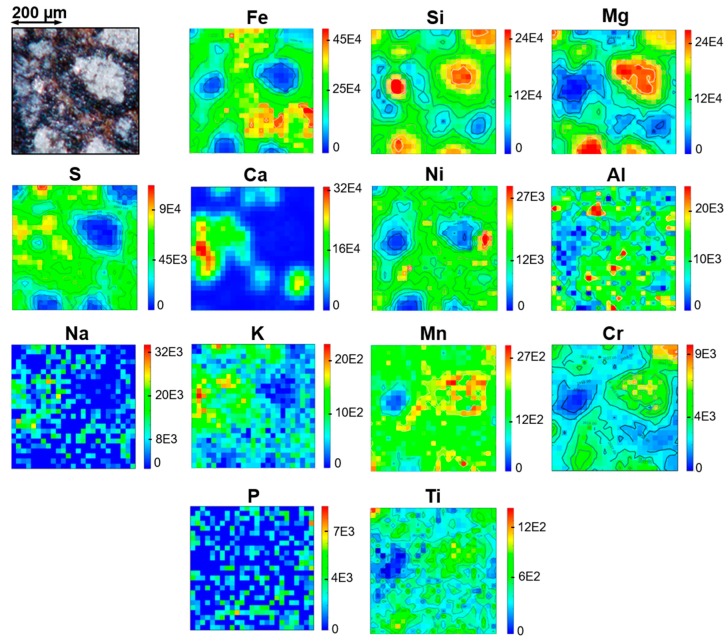
On the top left, the optical image of area I, then Fe, Si, Mg, S, Ca, Ni, Al, Na, K, Mn, Cr, P, and Ti maps obtained by micro-PIXE measurements on that area. The color scales are linear and directly indicate the elements’ concentrations in µg/g.

**Figure 10 life-09-00044-f010:**
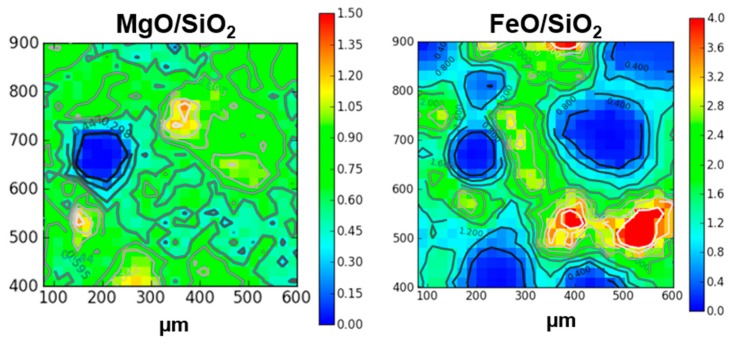
MgO/SiO_2_ and FeO/SiO_2_ ratio mappings deduced from micro-PIXE measurements on area I. For guidance, enstatite and forsterite compositions would correspond to a MgO/SiO_2_ ratio of 0.67 and 1.34, respectively. A fayalite-like composition would correspond to a FeO/SiO_2_ ratio of 2.4 (green on the right panel). Note the red colored areas on the right panel corresponding to iron-rich and silicon-poor regions.

**Figure 11 life-09-00044-f011:**
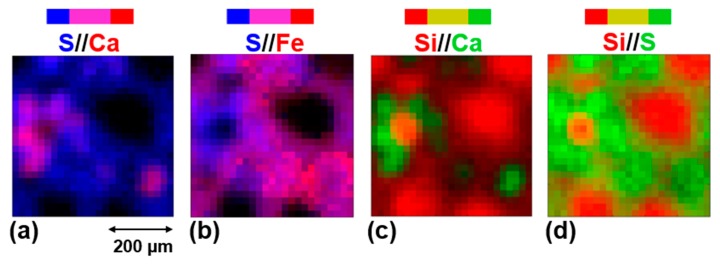
Micro-PIXE images overlaying different element mappings. Sulfur (blue) and calcium (red) are mapped in image (**a**), sulfur (blue) and iron (red) are mapped in image (**b**), silicon (red) and calcium (green) are mapped in image (**c**), and silicon (red) and sulfur (green) are mapped in image (**d**). Pink areas correspond to a co-localization of sulfur with either Ca (image (**a**)) or Fe (image (**b**)). Green-yellow hues correspond to a co-localization of silicon with either Ca (image (**c**)) or S (image (**d**)).

**Figure 12 life-09-00044-f012:**
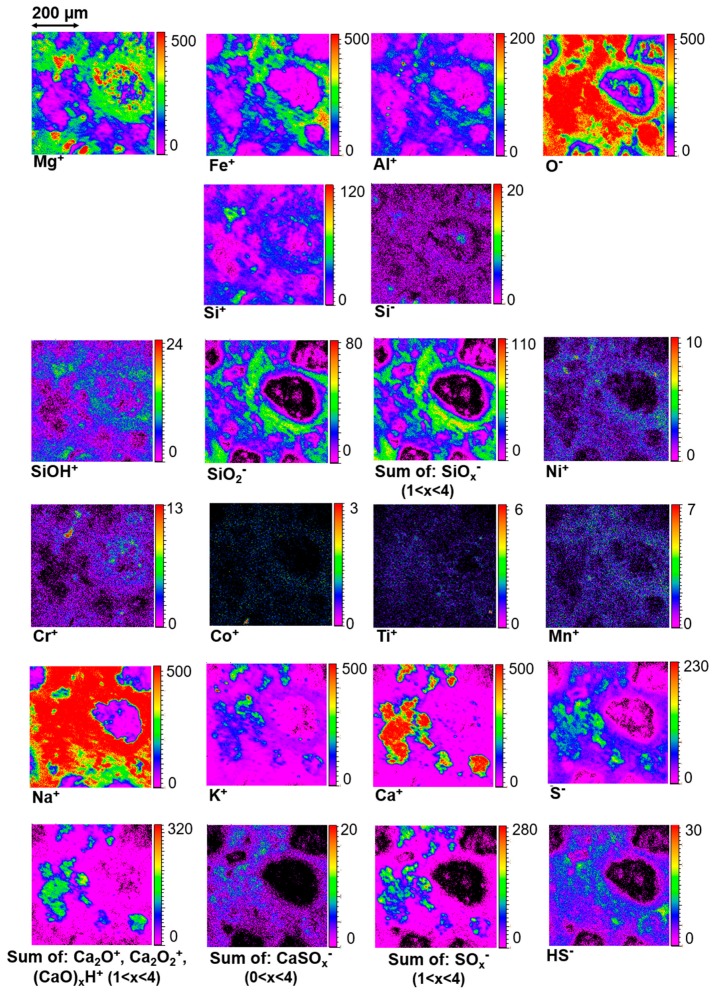
TOF-SIMS (Time-Of-Flight Secondary Ion Mass Spectrometry) mappings of different ions emitted from area I (500 × 500 µm²): Mg^+^, Fe^+^, Al^+^, O^−^, Si^+^, Si^−^, SiOH^+^, and SiO_2_^−^, the sum of silicon oxides anions SiO_x_^−^, Ni^+^, Cr^+^, Co^+^, Ti^+^, Mn^+^, Na^+^, K^+^, Ca^+^,and S^−^, calcium oxide and hydroxide cations’ sum, calcium sulfide and calcium sulfuroxide anions’ sum, sulfur oxides’ sum, and sulfur hydride anion. The various ionic silicon compounds are emitted from different regions of the meteorite indicating different minerals’ attributions that can be related to the metals showed in this Figure. The color scales indicate ion emission intensities in number of counts in the mass spectra.

**Figure 13 life-09-00044-f013:**
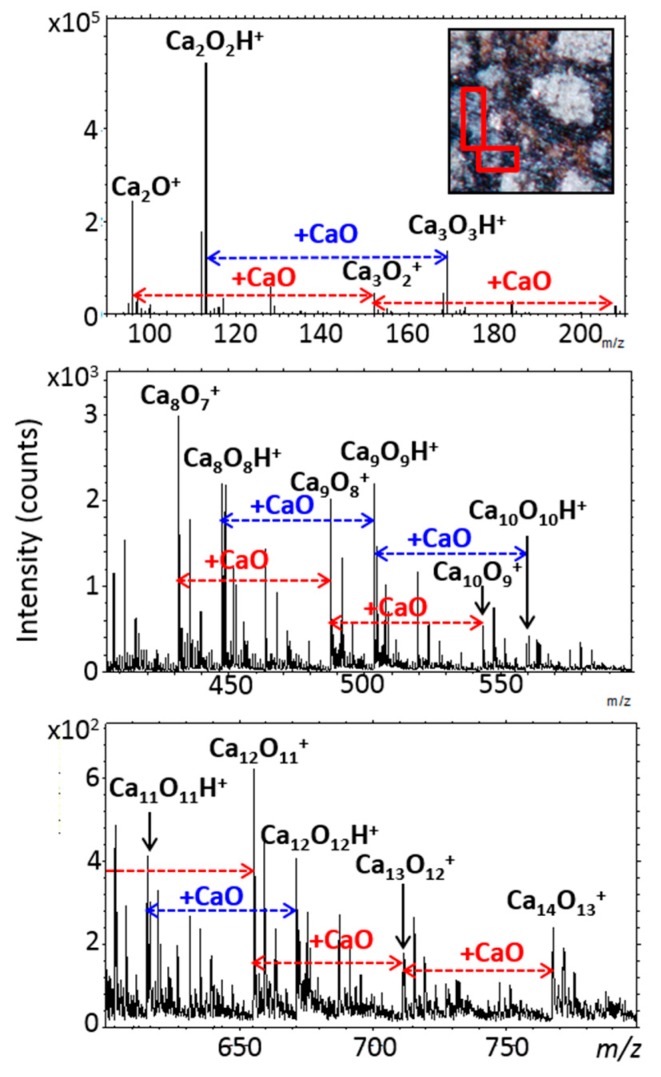
Positive emission spectra from the rectangular red areas on the optical image on the top. The mass spectra show the calcium oxide and hydroxide patterns between m/z 90 and 800, typical of a calcium carbonate and sulfate signatures.

**Figure 14 life-09-00044-f014:**
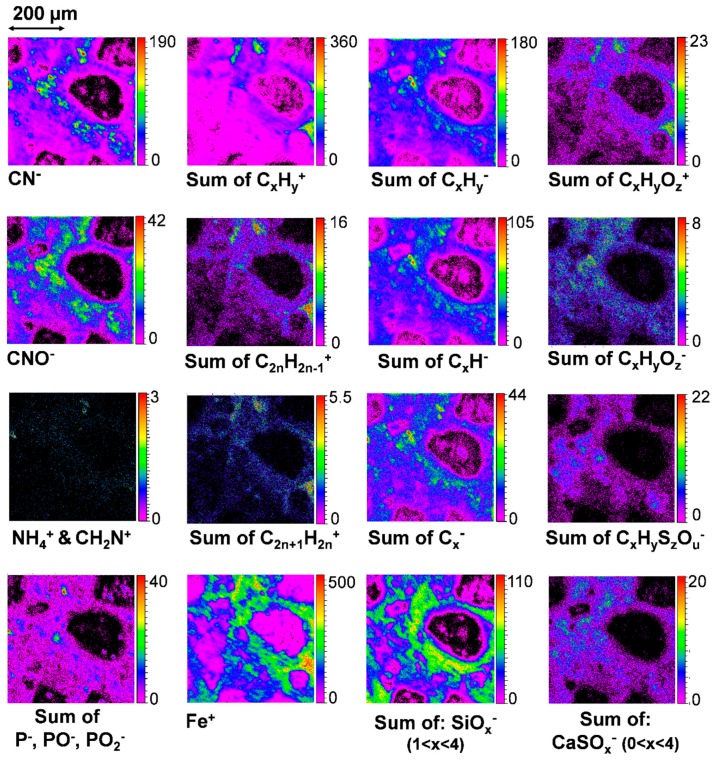
TOF-SIMS distribution images of different organic ions [CN^−^, CNO^−^, CH_2_N^+^, sum of C_x_H_y_^+^ (1 < x<21 and 1 < y < 15), sum of C_2n_H_2n–1_^+^ (2 < n < 16), sum of C_2n+1_H_2n_^+^ (3 < n < 15), sum of C_x_H_y_^−^ (1 < x < 21 and 1 < y < 9), sum of C_x_H^−^ (1 < x < 15), sum of C_x_^−^ (1 < x < 13), sum of C_x_H_y_O_z_^+^ (1 < x < 10, 1 < y < 15, 1 < z < 6), sum of C_x_H_y_O_z_^−^ (3 < x < 7, 1 < y < 6, 1 < z < 5), and sum of C_x_H_y_S_z_O_u_^−^ (2 < x < 3, 1 < y < 9, 1 < z < 2, 0 < u < 1)] compared to the distribution images of a few inorganic ions: P^−^ and its oxides, Fe^+^, sum of SiO_x_^−^ (1 < x < 4), and sum of CaSO_x_^−^ (0 < x < 4). (The color scales indicate ion emission intensities in number of counts in the mass spectra.)

**Figure 15 life-09-00044-f015:**
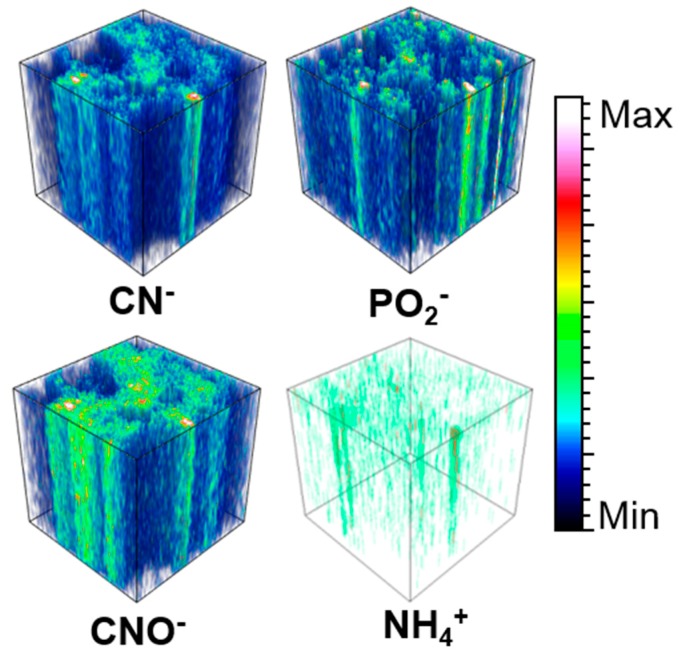
TOF-SIMS 3D plots of CN^−^, CNO^−^, PO_2_^−^, and NH_4_^+^. The vertical z axis corresponds to the depth according to the irradiation time with the argon clusters beam (Ar_1000_ at 10 keV).

**Figure 16 life-09-00044-f016:**
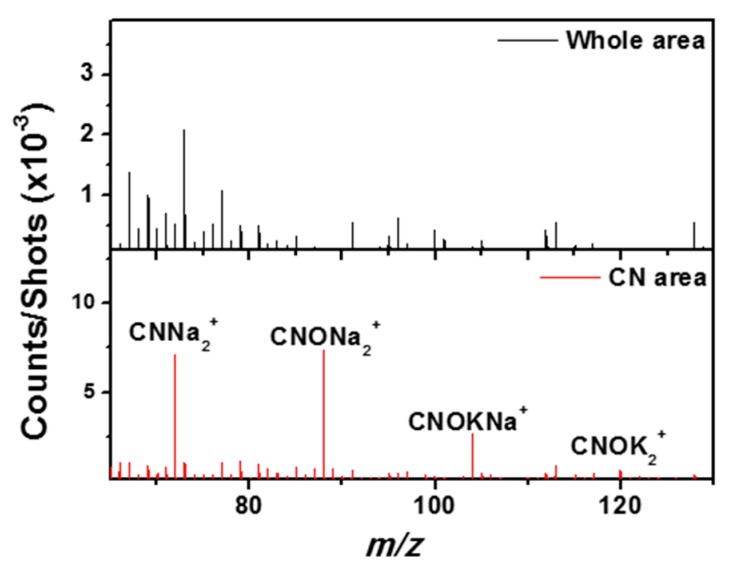
Area II mass spectra of positive ions between m/z 65 and 130. The black and red spectra correspond respectively to the whole area II and to the CN^-^-rich regions in that area. The K, Na-CN fragments’ emission is improved by the region of interest (ROI) selection.

**Figure 17 life-09-00044-f017:**
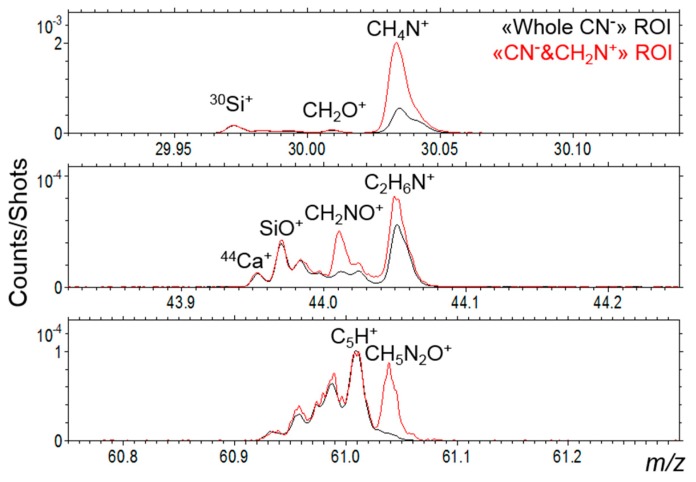
Area II mass spectra of positive ions emitted between m/z 29.9 and 61.3. The black and red spectra correspond respectively to the whole CN^−^-rich regions and the “CN^−^ and CH_2_N^+^” ROI selection. The emission of specific organic ions is revealed by the multi-correlation selection.

**Figure 18 life-09-00044-f018:**
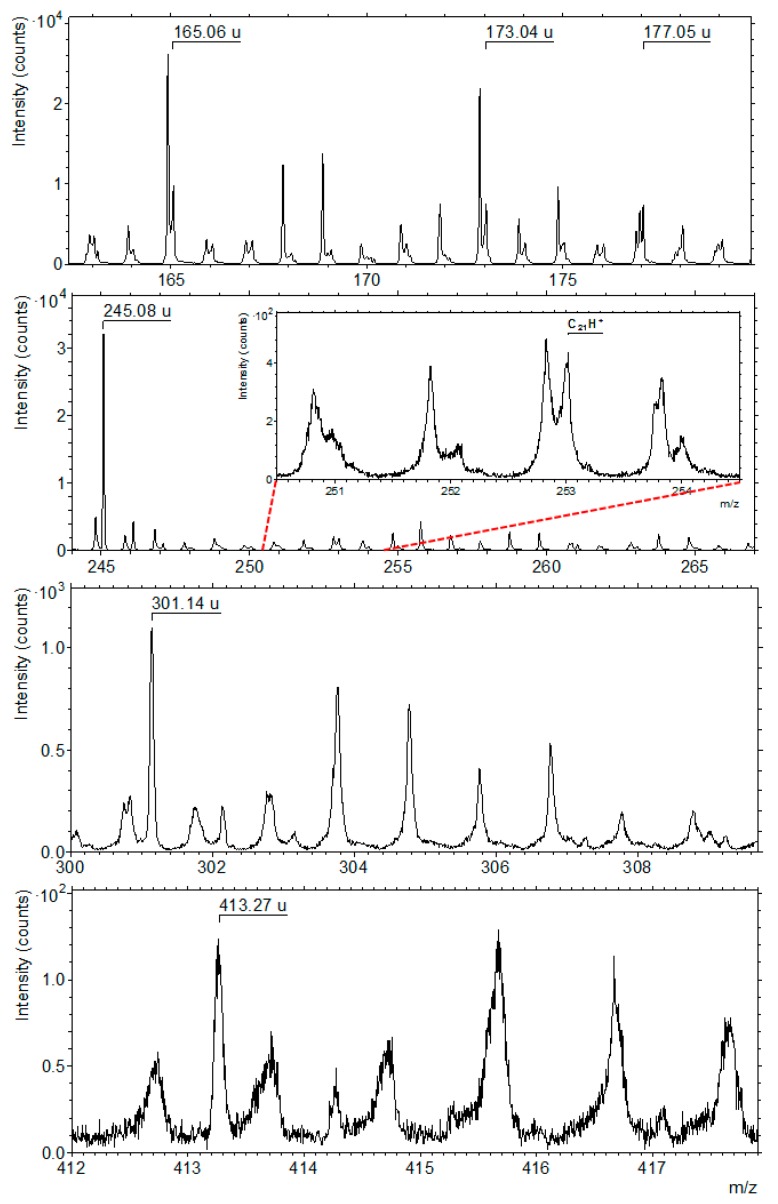
Successive zooms on the mass spectra of C_x_H_y_^+^-rich ROI in area II. Peaks corresponding to inorganic species are somewhat below the nominal mass due to the mass defect of the involved nucleotides. Peaks shifted right of the integer mass necessarily correspond to organic species due to their hydrogen content.

**Figure 19 life-09-00044-f019:**
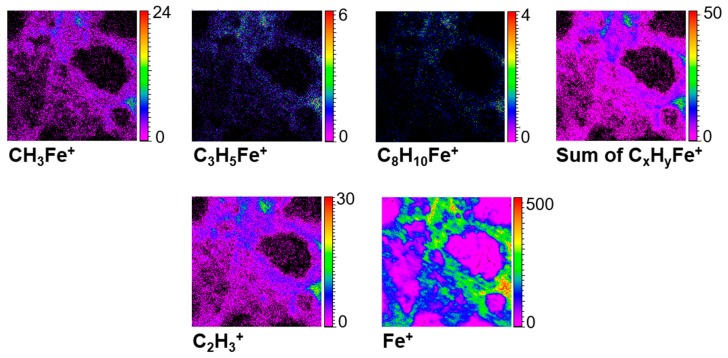
TOF-SIMS distribution images of a few organo-iron cations compared to C_2_H_3_^+^ and Fe^+^ distributions. The color scales indicate ion emission intensities in number of counts in the mass spectra.

**Figure 20 life-09-00044-f020:**
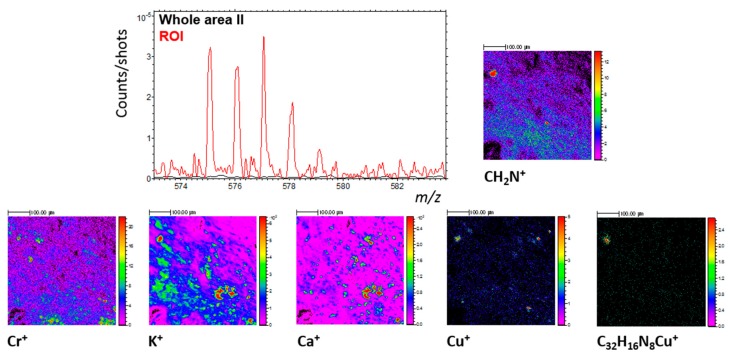
The top panel shows area II mass spectra of positive ions emitted between 570 and 585 u. The black and red spectra correspond respectively to the whole area II and the selected CH_2_N^+^-rich ROI. TOF-SIMS spatial distribution of CH_2_N^+^, Cr^+^, K^+^, Ca^+^, Cu^+^, and [C_32_H_16_N_8_Cu]^+^ (m/z 575.07) ions in area II are also shown. The organic emission is enhanced by the ROI selection and the group of peaks at 575 u are mainly attributed to the compound [C_32_H_16_N_8_Cu]^+^ (In the whole CH_2_N^+^-rich ROI, another organic compound is contributing to the peaks at m/z 576.08 and 577.08). As shown by the mappings, this signature is mainly emitted from the Cu^+^-rich spot in the top left of area II. This specific spot is also rich in Cr^+^, K^+^, and Ca^+^.

**Figure 21 life-09-00044-f021:**
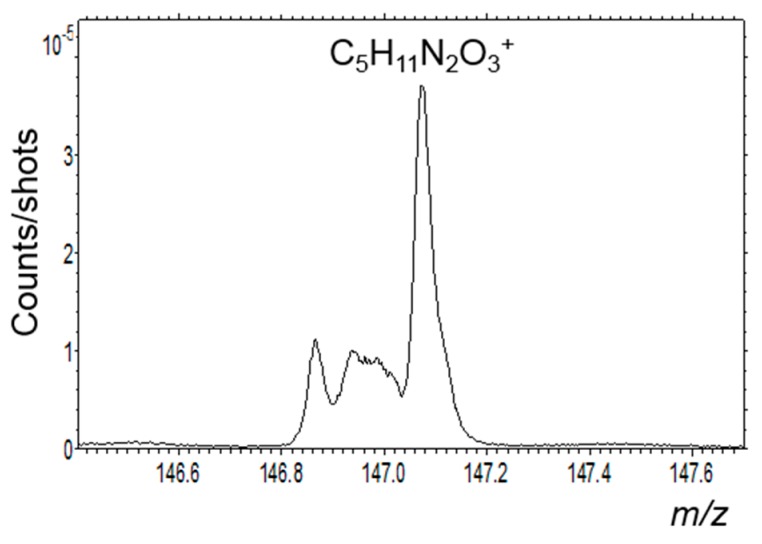
Area I mass spectrum of positive ions emitted from the C_x_H_y_^+^ ROI around mass 147 u and attributed to glutamine [M+H]^+^ molecular ion, [C_5_H_10_N_2_O_3_+H]^+^, with probable contribution of C_8_H_9_N_3_^+^ ion.

**Figure 22 life-09-00044-f022:**
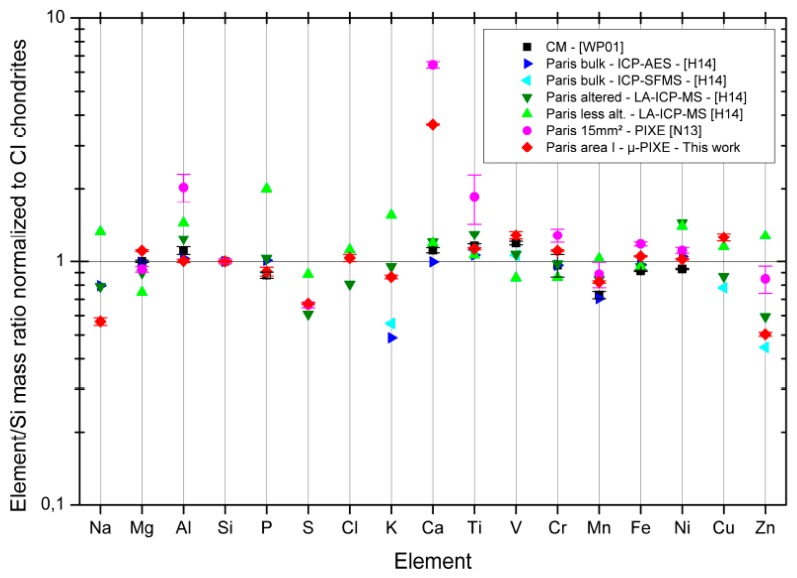
Elements to Si mass ratios normalized to CI chondrites [80]: (red diamond) the results obtained with micro-PIXE on area I and its surroundings (this work); (pink dots) the results obtained with PIXE technique on the whole millimetric Paris fragment [34]; (dark blue triangle) results obtained for Paris “bulk” with inductively coupled plasma associated with absorption emission spectrometry (ICP-AES) (after acid dissolution) [33]; (light blue triangle) results obtained for Paris “bulk” with ICP-SFMS (after acid dissolution) [33]; (dark green triangle) results obtained on a polished section of a 0.25 cm² altered region of Paris meteorite measured with laser ablation ICP-MS (mass spectrometry) [33]; (light green triangle) results obtained on 2 spots of a less altered region of Paris meteorite measured with laser ablation ICP-MS [33]; (black square) average composition of CM chondrites based on Wolf and Palme [81] results.

**Figure 23 life-09-00044-f023:**
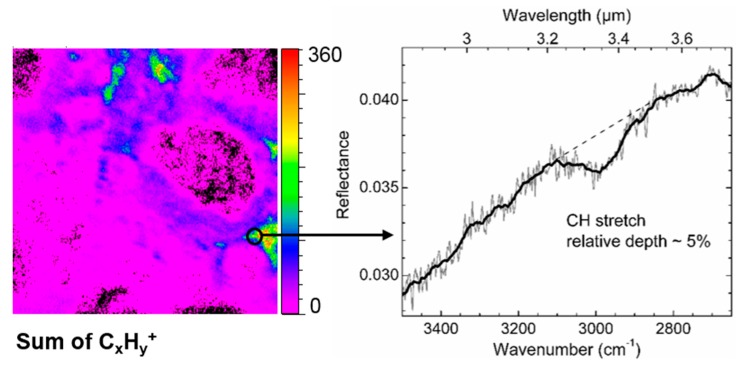
On the left, the spatial distribution of C_x_H_y_^+^ ions in area I. On the right, an example of C–H signature obtained by FTIR micro-reflectance in region C of area I (Figure 2). The black circle on the left image shows precisely where the IR spectrum was measured.

**Table 1 life-09-00044-t001:** Elemental concentrations (in ppm, µg/g) for area I with the limit of detection (LOD) and the errors in percentages.

	Concentration (ppm)	LOD (ppm)	Fit Error (%)	Stat. Error (%)
Na	3450	139	3.18	2.82
Mg	12,9103	83	0.44	0.12
Al	10,395	59	1.12	0.49
Si	130,313	53	0.4	0.08
P	1070	59	3.72	3.27
S	43,764	13	0.54	0.13
Cl	878	29	2.98	2.55
K	572	9	1.3	0.97
Ca	41,109	11	0.24	0.03
Ti	619	12	0.64	1.39
V	85	8	3.4	4.49
Cr	3573	7	0.34	0.14
Mn	1935	35	0.75	0.85
Fe	236,305	11	0.17	0.01
Ni	13,424	6	0.26	0.07
Cu	201	12	3.27	6.43
Zn	198	3	1.41	1.11

**Table 2 life-09-00044-t002:** List of the most intense mass peaks appearing in positive and negative modes in the CxHy^±^-rich ROI and corresponding to heteroatoms-containing organic formulas. * The numbers between brackets indicate the deviation to the m/z value of the detected peak.

Positive Mode		Negative Mode	
Detected m/z	Proposed Attribution *	Detected m/z	Proposed Attribution *
43.892[0]	CHP^+^ (−1.3 ppm)	40.019[3]	C_2_H_2_N^−^ (0.4 ppm)
60.011[9]	C_2_H_5_P^+^ (−6.7 ppm)	43.019[5]	C_2_H_3_O^−^ (13.4 ppm)
68.989[4]	C_3_H_2_P^+^ (7.5 ppm)	45.034[2]	C_2_H_5_O^−^ (−8.8 ppm)
73.016[2]	C_2_H_3_NO_2_^+^ (5.5 ppm)	55.018[1]	C_3_H_3_O^−^ (−15 ppm)
73.054[5]	C_2_H_7_N_3_^+^ (−5.0 ppm)	56.980[7]	C_2_HS^−^ (5.2 ppm)
	C_4_H_9_O^+^ (−18.2 ppm)		SiCHO^−^ (9.2 ppm)
74.023[9]	C_2_H_4_NO_2_^+^ (2.0 ppm)	59.013[2]	C_2_H_3_O_2_^−^ (−11 ppm)
75.032[8]	C_2_H_5_NO_2_^+^ (8.1 ppm)	67.017[5]	C_2_HN_3_^−^ (−1.2 ppm)
77.001[4]	C_5_HO^+^ (−10.8 ppm)	69.989[0]	*H_3_SOF^−^ (−5.6 ppm)* †
83.046[7]	C_3_H_5_N_3_^+^ (−17.4 ppm)		C_3_H_2_S^−^ (10.7 ppm)
89.043[6]	C_4_H_9_S^+^ (2.8 ppm)	71.012[7]	C_3_H_3_O_2_^−^ (−16.3 ppm)
101.047[3]	C_4_H_7_NO_2_^+^ (2.1 ppm)		CH_2_F_3_^−^ (18.1 ppm)
102.053[4]	C_5_H_10_S^+^ (13.9 ppm)	71.050[8]	C_2_H_5_N_3_^−^ (−16.5 ppm)
106.085[3]	C_4_H_12_NO_2_^+^ (−9.2 ppm)		C_4_H_7_O^−^ (8.4 ppm)
108.101[3]	C_2_H_12_N_4_O^+^ (6.5 ppm)	74.989[7]	C_2_H_3_SO^−^ (−10.4 ppm)
112.009[3]	C_4_H_4_SN_2_^+^ (3 ppm)		*C_2_O_2_F*^−^*(12 ppm)* †
	SiC_3_H_4_N_2_O^+^ (5.1 ppm)	76.029[0]	CH_4_N_2_O_2_^−^ (15 ppm)
113.007[9]	C_5_H_5_SO^+^ (20.6 ppm)	87.020[4]	C_2_H_3_N_2_O_2_^−^ (4.2 ppm)
120.986[1]	C_5_HSN_2_^+^ (5.3 ppm)	90.040[2]	C_3_H_8_SN^−^ (17.2 ppm)
121.120[3]	C_4_H_15_N_3_O^+^ (−5.5 ppm)		SiC_2_H_8_NO^−^ (20.1 ppm)
122.126[5]	C_4_H_16_N_3_O^+^ (18.9 ppm)	91.018[3]	C_4_HN_3_^−^ (−0.8 ppm)
131.096[6]	C_6_H_13_NO_2_^+^ (18.9 ppm)		C_6_H_3_O^−^ (−15.6 ppm)
146.070[4]	C_5_H_10_N_2_O_3_^+^ (12.1 ppm)	93.033[0]	C_6_H_3_O^−^ (−2.2 ppm)
165.055[8]	C_9_H_9_O_3_^+^ (7.1 ppm)		C_6_H_5_O^−^ (−16.7 ppm)
173.044[0]	C_7_H_9_O_5_^+^ (−2.6 ppm)	98.007[3]	C_4_H_4_SN^−^ (3.3 ppm)
177.055[5]	C_10_H_9_O_3_^+^ (5.1 ppm)	104.024[0]	C_5_H_2_N_3_^−^ (−7.9 ppm)
245.078[8]	C_14_H_13_O_4_^+^ (−8.4 ppm)		C_2_H_4_N_2_O_3_^−^ (12.5 ppm)
		108.001[9]	CH_4_SN_2_O_2_^−^ (18.5 ppm)
			C_6_H_4_S^−^ (−18.7 ppm)
		121.025[5]	C_2_H_5_N_2_O_4_^−^ (0.4 ppm)
			C_5_H_3_N_3_O^−^ (−18.7 ppm)
		132.001[2]	C_3_H_4_SN_2_O_2_^−^ (10.1 ppm)
			C_8_H_4_S^−^ (−20.4 ppm)

† Both ions indicated here in italics are possible attributions for the corresponding mass peaks, but they are probably not originating from an organic compound.

**Table 3 life-09-00044-t003:** High mass fragments checked in negative mode to look for amino acids. A green tick indicates an identified fragment, whereas a red tick indicates an attribution with an error of tens of ppm. The amino acids are nominated by their 3-letter abbreviation; “Orn” being ornithine, a non-proteinogenic diamino acid.

	Ala	Phe	Asp	Ile	Asn	His	Leu	Lys	Met	Orn	Pro	Ser	Thr	Trp	Tyr	Val
NH^−^	✓		✓	✓	✓	✓	✓	✓	✓	✓	✓	✓	✓	✓	✓	✓
NH_2_^−^	✓		✓		✓	✓	✓	✓	✓		✓	✓		✓	✓	✓
CN^−^		✓	✓		✓	✓			✓			✓				✓
CH_2_N^−^				✓				✓		✓	✓		✓		✓	
NO^−^		✓	✓		✓	✓						✓				
S^−^									✓							
HS^−^									✓							
CHS^−^									✓							
CH_3_S^−^									✓							
C_2_N^−^						✓		✓			✓				✓	
C_2_HN^−^	✓		✓	✓	✓	✓	✓	✓	✓	✓	✓	✓	✓	✓	✓	✓
C_2_H_2_N^−^		✓														
C_2_O^−^	✓			✓		✓	✓	✓	✓	✓	✓	✓	✓	✓	✓	✓
C_2_HO^−^			✓		✓								✓			
CN_2_^−^						✓										
C_2_H_2_N^−^	✓		✓	✓	✓	✓	✓	✓	✓	✓	✓	✓		✓	✓	✓
CNO^−^	✓		✓	✓	✓	✓	✓	✓	✓	✓	✓	✓	✓	✓	✓	✓
CO_2_^−^	✓			✓	✓	✓	✓	✓	✓	✓	✓	✓		✓	✓	✓
CH_2_NO^−^						✓						✓				
CHO_2_^−^	✓	✓	✓	✓	✓	✓	✓	✓	✓	✓	✓	✓	✓	✓	✓	✓
NO_2_^−^		✓														
C_4_H^−^							✓							✓		
C_3_N^−^						✓								✓		
C_2_H_2_O_2_^−^			✓		✓											
C_2_H_3_O_2_^−^			✓		✓											
C_2_H_4_NO^−^					✓							✓				
C_3_H_4_NO^−^			✓		✓											
C_3_H_3_O_2_^−^	✓		✓	✓		✓	✓	✓	✓	✓	✓	✓	✓	✓	✓	✓
C_2_H_2_NO_2_^−^	✓		✓	✓	✓	✓	✓	✓	✓	✓	✓	✓	✓	✓	✓	✓
C_3_H_2_NO_2_^−^					✓											
C_3_H_4_NO_2_^−^	✓				✓			✓								
C_3_H_6_NO_3_^−^												✓				
C_3_H_7_NO_3_^−^												✓				
C_4_H_5_N_2_O_2_^−^					✓											
C_4_H_4_NO_3_^−^					✓											
C_5_H_6_NO_2_^−^		✓														
C_5_H_8_NO_2_^−^											✓					
C_4_H_8_NO_3_^−^													✓			
C_4_H_9_NO_3_^−^													✓			
C_8_H_7_O^−^															✓	
C_4_H_6_NO_4_^−^			✓													
C_6_H_11_N_2_O_2_^−^								✓								

**Table 4 life-09-00044-t004:** High mass fragments checked in positive mode to look for amino acids. The green tick indicates the fragments identified and precisely attributed. The amino acids are nominated by their 3-letter abbreviation, “Orn” being ornithine, a non-proteinogenic diamino acid.

	Ala	Phe	Asp	Ile	Asn	His	Leu	Lys	Met	Orn	Pro	Ser	Thr	Trp	Tyr	Val
NH_3_^+^		✓	✓	✓	✓	✓	✓	✓	✓	✓	✓	✓	✓	✓	✓	✓
NH_4_^+^	✓		✓	✓	✓	✓	✓	✓	✓	✓	✓	✓	✓	✓	✓	✓
C_2_H_3_^+^				✓												
CH_2_N^+^			✓	✓	✓	✓	✓	✓	✓	✓	✓	✓	✓	✓	✓	✓
CH_3_N^+^			✓	✓		✓										
C_2_H_5_^+^				✓			✓	✓	✓		✓		✓		✓	✓
CH_4_N^+^	✓		✓	✓	✓	✓	✓	✓	✓	✓	✓	✓	✓	✓	✓	✓
CH_5_N^+^										✓						
CH_3_O^+^												✓				
S^+^									✓							
C_3_H_3_^+^														✓		
C_2_H_2_N^+^			✓		✓	✓										
C_2_H_3_N^+^		✓	✓		✓	✓										
C_2_H_4_N^+^	✓	✓	✓	✓	✓	✓	✓	✓	✓	✓	✓	✓	✓	✓	✓	✓
C_2_H_5_N^+^			✓		✓	✓					✓	✓				
C_2_H_6_N^+^	✓		✓	✓	✓	✓	✓	✓	✓	✓	✓	✓	✓	✓	✓	✓
CH_2_NO^+^			✓		✓	✓										
CH_4_NO^+^			✓		✓	✓										
C_2_H_5_O^+^													✓			
CH_3_S^+^									✓							
C_4_H_3_^+^														✓		
C_3_H_2_N^+^			✓	✓	✓	✓										
C_4_H_5_^+^														✓		
C_3_H_4_N^+^			✓	✓	✓	✓	✓	✓	✓	✓	✓		✓			
C_3_H_6_N^+^			✓	✓	✓	✓	✓	✓	✓	✓	✓		✓	✓	✓	✓
C_3_H_8_N^+^			✓	✓	✓	✓	✓	✓	✓	✓			✓	✓	✓	✓
C_2_H_5_NO^+^					✓											
C_2_H_5_S^+^									✓							
C_3_HNO^+^					✓											
C_3_H_2_NO^+^					✓											
C_4_H_6_N^+^								✓			✓					
C_3_H_5_N_2_^+^						✓										
C_3_H_4_NO^+^			✓		✓											
C_4_H_8_N^+^				✓	✓		✓	✓	✓	✓	✓		✓			✓
C_3_H_6_NO^+^			✓	✓	✓								✓			
C_4_H_10_N^+^			✓	✓				✓								✓
C_3_H_7_NO^+^													✓	✓	✓	✓
C_3_H_8_NO^+^													✓			
C_2_H_5_NO_2_^+^			✓	✓	✓											
C_6_H_5_^+^		✓														
C_4_H_5_N_2_^+^						✓										
C_5_H_8_N^+^							✓	✓	✓	✓	✓		✓			
C_4_H_6_NO^+^					✓								✓			
C_5_H_10_N^+^					✓			✓								
C_5_H_12_N^+^				✓			✓	✓	✓	✓		✓	✓	✓	✓	✓
C_3_H_7_N_2_O^+^					✓											
C_3_H_8_NO_2_^+^	✓															
C_7_H_7_^+^		✓														
C_5_H_5_N_2_O^+^					✓											
C_8_H_7_N^+^														✓		
C_8_H_10_N^+^		✓														
C_9_H_8_N^+^														✓		
C_8_H_10_NO^+^		✓													✓	
C_5_H_13_N_2_O_2_^+^										✓						
C_4_H_8_NO_4_^+^			✓													
C_6_H_10_N_3_O_2_^+^						✓										
C_9_H_12_NO_2_^+^		✓														
C_4_H_8_N_2_O_3_K^+^					✓											
C_4_H_7_NO_4_K^+^			✓

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
