# Peer review of "A Mineralogical Context for the Organic Matter in the Paris Meteorite Determined by A Multi-Technique Analysis"

_life, 2019, doi:10.3390/life9020044_

Round 1
Reviewer 1 Report
The manuscript entitled “A mineralogical context for the organic matter in the Paris meteorite determined by a multi-technique analysis” (ID: life-485369) by Noun et al. described comprehensive analyses of the Paris meteorite (the least altered CM) using multi-technique measurements including "surface roughness measurement", "visible and near-IR diffuse reflectance spectroscopy", "mid-IR reflectance spectroscopy", "Raman micro-spectroscopy", "micro-PIXE" and "TOF-SIMS". The coordinate measurements revealed inorganic and organic matter distribution as well as mineral occurrence in the Paris meteorite. I have no objections that this paper will be published in life. However, I am not sure the possibility of amino acid(-moieties) identification only by this study. I cannot understand why the authors have chosen these 16 amino acids (Ala to Val) in Table 3. The very complex amino acids such as Trp and Try (Tyr?) have never been reported from extraterrestrial materials, while the most abundant meteoritic amino acids such as Gly and Glu were not included in Table 3. I recommend reconsideration of the possible amino acid occurrence in the meteorite. Specific comments are provided as follows.
Line 70: I prefer “chromatography” rather than “separation”, because the solvent extraction implies “separation”.
Line 76: “they were all applied after extraction”; microprobe approaches such as two-step laser mass spectrometry do NOT accompany “extraction”.
Line 99-100: Please explain about “RBS”. In addition, by Piani et al. (2018, Nature Astronomy), bulk carbon content of the Paris meteorite was reported as 1.64 wt%. I do not understand “a whole carbon content of 7.7 ± 1.0 at.%”.
Line 109-110: PAHs were extracted with dichloromethane/methanol in ref. [2], not using “either water or acid water”.
Line 110-111: I prefer “both the water extract and the acid hydrolysate” rather than “both the hydrolysate and the acid hydrolysate”. The “hydrolysate” implies “hydrolysis product”.
Line 163-164: Please describe briefly why the authors selected “Area I and Area II” for analysis of this study.
Line 278: “TOF-SIMS”: Please describe the mass range (m/z ??-???) for the measurement of this study.
Line 335-350: 3D images of Figure 3 were obtained before analysis? If possible, please show the 3D images AFTER a suite of analysis. It is very helpful to evaluate the alteration of the sample surface during the analyses.
Line 420-426: Please indicate the measuring locations of (a)-(e) on the sample surface of Area I (e.g. Panel A in Figure 4 or in Figure 8). The arrow of panel (c’ and f) in Figure 5 is slightly deviated from 2.75 mm (OH).
Line 471: “(Figure 24)” is correct?
Line 555: Please show the phosphorous (P) distribution in Figure 9. The P concentration is 1070 (ppm) in Table 1. The P distribution is important with relation to “Sum of P-, PO-, PO2-“ in Figure 15 and “PO2-“ in Figure 16 by TOF-SIMS.
Line 610-662: Figure 12 and Figure 13 should be combined into one figure.
Line 801-803: It is hard for me to read the ion distribution, especially for C3H5Fe+ and C8H10Fe+.
Line 834-835: “some amino acid standards have been analyzed with the ION-TOF mass spectrometer”; please specify the name of amino acids.
Line 871: Again I cannot understand why the authors have chosen these 16 amino acids (Ala to Val).
Line 1000: Isotope measurement of 15N/14N using CN- by NanoSIMS usually utilizes Cs+ as a primary beam. In this study, Bi3+ was used for as a primary beam, which may have a different process to produce CN-. Please justify the similar effect for ionization, or give a reference.
Author Response
Response to Reviewer 1 Comments
In black are the reviewer’s comments and questions. Our responses are in red. The highlighted sentences indicate the modifications added to the manuscript. All the line numbers, figure numbers, table numbers etc. we refer to in these answers correspond to the 1st version of the manuscript. Of course these numbers may change slightly in the revised version due to the modifications added.
The manuscript entitled “A mineralogical context for the organic matter in the Paris meteorite determined by a multi-technique analysis” (ID: life-485369) by Noun et al. described comprehensive analyses of the Paris meteorite (the least altered CM) using multi-technique measurements including "surface roughness measurement", "visible and near-IR diffuse reflectance spectroscopy", "mid-IR reflectance spectroscopy", "Raman micro-spectroscopy", "micro-PIXE" and "TOF-SIMS". The coordinate measurements revealed inorganic and organic matter distribution as well as mineral occurrence in the Paris meteorite. I have no objections that this paper will be published in life. However, I am not sure the possibility of amino acid(-moieties) identification only by this study. I cannot understand why the authors have chosen these 16 amino acids (Ala to Val) in Table 3. The very complex amino acids such as Trp and Try (Tyr?) have never been reported from extraterrestrial materials, while the most abundant meteoritic amino acids such as Gly and Glu were not included in Table 3. I recommend reconsideration of the possible amino acid occurrence in the meteorite. Specific comments are provided as follows.
The answer to the comment about amino acids identification is below, following another related comment of the reviewer.
Line 70: I prefer “chromatography” rather than “separation”, because the solvent extraction implies “separation”.
We clarified the sentence as suggested by the reviewer.
Line 76: “they were all applied after extraction”; microprobe approaches such as two-step laser mass spectrometry do NOT accompany “extraction”.
This is true. Thank you for spotting the misleading generalization of the sentence. We modified the sentence accordingly.
Line 99-100: Please explain about “RBS”. In addition, by Piani et al. (2018, Nature Astronomy), bulk carbon content of the Paris meteorite was reported as 1.64 wt%. I do not understand “a whole carbon content of 7.7 ± 1.0 at.%”.
The set up on a 1.7 MV tandem accelerator applied for the RBS measurements and the simulation code used to determine the carbon elemental content of the Paris meteorite sample measured are all described in Noun et al. 2013. We re-mentioned the reference at line 99-100 to avoid confusion.
We were not aware of the value reported by Piani et al. It is of course extremely interesting to add this other measurement of the carbon content of the Paris meteorite. It is however, impossible to properly compare both values, as one is in atomic percentage, whether the other is in weight percentage. A correct conversion needs the whole elemental composition of the studied sample (or at least, its composition in major elements including oxygen) in atomic percentage for the RBS value, or in weight percentage for Piani et al. value. Unfortunately, we do not have the content in Si, Fe, Mg and other major elements in atomic percentage, and we do not have the oxygen content in weight percentage. Of course, it is expected to have a value of carbon weight percentage lower than the carbon atomic percentage value. Indeed, if we make the calculation for the CI chondrite reference composition (Lodders, Palme and Gail, 2009 ; Palme, Lodders and Jones, 2014), we will find a content in carbon of 3.5 wt.% and 4.5 at.%. Besides, the carbon content of different CM chondrites was measured by Pearson et al. 2006, for example. They found values ranging from 2.16 to 4.05 wt.% ; the mean value being 2.68 wt.%. Compared to these latter values, Piani et al. value of 1.64 wt.% for the Paris meteorite seems to be a bit low, even though no error bars or uncertainties on the value are given in the paper to possibly moderate this impression. On the other hand, the value of 7.7 ± 1.0 at.% determined by Noun et al. 2013 seems to be high compared to the values found for other CM chondrites.
We added the Piani et al. reference and the value they found (line 99-100), and we adjoined a few comments on both measured values as compared to Pearson et al. 2006.
Line 109-110: PAHs were extracted with dichloromethane/methanol in ref. [2], not using “either water or acid water”.
Yes, we corrected this sentence.
Line 110-111: I prefer “both the water extract and the acid hydrolysate” rather than “both the hydrolysate and the acid hydrolysate”. The “hydrolysate” implies “hydrolysis product”.
We modified the wording accordingly.
Line 163-164: Please describe briefly why the authors selected “Area I and Area II” for analysis of this study.
There is no particular reason for choosing those both areas except what is mentioned at lines 165-167 about choosing an area containing chondrules and matrix regions for area I, and then an area dominated by matrix for area II to be able to focus on the organic content measured by TOF-SIMS. A part from that, it was a random choice.
Line 278: “TOF-SIMS”: Please describe the mass range (m/z ??-???) for the measurement of this study.
We added this information to section 2.6. For the TOF-SIMS measurements, the mass range was 1-3000 u and in the analyses spectra, we detected peaks from m/z 1 up to 1000 u.
Line 335-350: 3D images of Figure 3 were obtained before analysis? If possible, please show the 3D images AFTER a suite of analysis. It is very helpful to evaluate the alteration of the sample surface during the analyses.
As mentioned in section 2.1, lines 185-187, these roughness measurements were performed after the whole suite of analyses to be sure that the surface roughness of the measured areas and of the matrix regions especially, were low enough to not induce biases in TOF-SIMS analyses and results’ interpretations particularly. We added a sentence at the end of section 3.1 to insist on this important outcome. We did not apply the 3D laser scanning measurement before the sequence of analyses, because there was a risk that the laser beam could damage or alter the organic content of the meteorite which is of major interest in this study.
Line 420-426: Please indicate the measuring locations of (a)-(e) on the sample surface of Area I (e.g. Panel A in Figure 4 or in Figure 8). The arrow of panel (c’ and f) in Figure 5 is slightly deviated from 2.75 mm (OH).
We modified Figure 2 to add on it the location of the spectra a-e shown in Figure 5.
The arrows for spectrum f in Figure 5 point to the iron hydroxide bands contributing to the whole spectrum. Its O-H vibration is situated at the right shoulder of the 2.75 µm band, at ~3.1 µm as pointed by the arrow and indicated in the text, line 415. We added some clarification in the figure caption to avoid confusion.
Line 471: “(Figure 24)” is correct?
Yes, we refer here to the right panel on Figure 24. We added the detail to avoid confusion.
Line 555: Please show the phosphorous (P) distribution in Figure 9. The P concentration is 1070 (ppm) in Table 1. The P distribution is important with relation to “Sum of P-, PO-, PO2-“ in Figure 15 and “PO2-“ in Figure 16 by TOF-SIMS.
We added the phosphorous distribution to Figure 9. As presented in Table 1, the whole content in phosphorous is quite low (1070 ppm) compared to the other elements measured by micro-PIXE and its limit of detection quite high (59 ppm) compared to other minor elements such as Ti (12 ppm) or Cr (7 ppm). Together with the large pixel size (10-20 µm) of micro-PIXE, these constraints cannot give a very precise and interesting mapping for phosphorous compared to TOF-SIMS mappings. Indeed, these latter have a pixel size of less than 2µm and they benefit from the high ionization yield of POx- species.
Line 610-662: Figure 12 and Figure 13 should be combined into one figure.
We merged both figures.
[Note to editor: Please note that this modification also changes the total number of figures and the references to them in the text, in the new revised version of the manuscript.]
Line 801-803: It is hard for me to read the ion distribution, especially for C3H5Fe+ and C8H10Fe+.
We improved the color scales on those panels and enhanced the color to make them more clear.
Line 834-835: “some amino acid standards have been analyzed with the ION-TOF mass spectrometer”; please specify the name of amino acids.
We clarified this sentence in the manuscript. The amino acid standards that have been analyzed are all the ones that are presented and assessed in Tables 3 and 4. We also corrected the misprint for Tyrosine symbol (Tyr and not “Try”) in both tables 3 and 4.
Line 871: Again I cannot understand why the authors have chosen these 16 amino acids (Ala to Val).
As clarified above, those were the 16 amino acids that we could assess thanks to the reference measurements on the corresponding standards. As stated line 854, the only amino acids that we unambiguously identify thanks to this study are alanine, asparagine and aspartic acid. Glutamine is strongly suspected (lines 855-856), whereas the peaks found for all the other amino acids assessed and presented in table 3 and 4 are not enough to have a firm identification of those molecules as species present in the matrix of the analyzed sample. We modified the sentences at line 853-860 to be clearer and avoid any misleading conclusion here.
Line 1000: Isotope measurement of 15N/14N using CN- by NanoSIMS usually utilizes Cs+ as a primary beam. In this study, Bi3+ was used for as a primary beam, which may have a different process to produce CN-. Please justify the similar effect for ionization, or give a reference.
The secondary ion emission processes are not different, however the doses applied are. In NanoSIMS measurements the primary ion doses are much higher than the ones we apply in TOF-SIMS. Typically, in NanoSIMS experiments total doses higher than 1015 Cs+/cm² are applied to increase as much as possible the ionization yield of the mass peaks investigated to determine isotopic ratios (for example 12C14N- and 12C15N-). In those experiments, Cs+ ions are purposely implanted on the surface, as much secondary ions as possible are extracted from the surface, and this latter is knowingly modified by this harsh treatment. For comparison, the TOF-SIMS total dose we applied is 5x1011 ions/cm² for area I and is 1x1012 ions/cm² for area II (line 282).
Moreover, the CN- ion peak is used in NanoSIMS measurements for nitrogen isotopic studies because it is very efficiently emitted in all SIMS measurements (with diverse primary beams) and because in most of these studies that ion was logically thought to be related to the organic content of the sample. However, if a sample containing a cyanide salt and some other nitrogen containing organic components is analyzed by a NanoSIMS beam focusing on the CN- mass range, CN- ions from the salt components will necessarily and easily be emitted together with the ones emitted from the other components. If the salt has a different isotopic composition from that of the other organic compounds, the difference would be reflected in the NanoSIMS measurements by showing different isotopic ratios. However, only complementary experiments (such as TOF-SIMS measurements) could determine how the isotopic differences are related to different chemical compositions. In our TOF-SIMS study, we needed the whole mass spectrum and thus, other mass peaks, to determine the two different chemical signatures responsible for the CN- ions detected.

Reviewer 2 Report
This manuscript is a detailed technical description of the analytical and interpretation work performed in characterising a carbonaceous chondrite. This is of high value in establishing good practices in interpretation and very informative in making direct comparisons between techniques for the same purpose, especially when combining remote investigations with microscopic, detailed in situ analyses. This is interesting when the composition of the smaller bodies of our solar system is compared with meteorites, and this work adds to past analyses with other techniques, differentiating in some respect with them on characterising a totally untreated sample of the meteorite.
The manuscript uses very good English and no typos have been detected. In places, the text is too descriptive and might confuse the reader, but it is the technical aspect of the paper that causes this effect. It is not typical bold characters to be used for whole sentences, so this feature might have to be removed. Also, smaller fonts in captions should be made equal to the rest of the text of the caption. Finally, sentences in brackets should be removed and be treated as main text. Title and abstract reflect the text.
This paper is suggested to be published with minor modifications and prior to some clarifications. Some points to be considered are the following:
Paragraph 1 (Introduction):
In the introduction there is a list of references covering major techniques employed in characterising the organic matter of carbonaceous chondrites. The authors could also consider together with reference [19] to mention the following reference which is along the same lines, however describing a different setup with a special use of a new, segmented ion trap:
· Sabbah et al, 2017, Identification of PAH Isomeric Structure in Cosmic Dust Analogues: the AROMA setup, Astrophys J. 2017 Jul 1; 843(1): 34. doi: 10.3847/1538-4357/aa73dd
Also, in lines 152-155, where references are given to the Stardust mission, of interest might be the reference doing the same analysis with TOF-SIMS but rather using C60 as an ion beam, optimum for heavy organics compounds. Please, look at the following:
· Rost et al, 2008, C60 ToF-SIMS: A tool for high-resolution mapping of elements and organic compounds. Geochimica et Cosmochimica Acta 72(12).
Lines 180-187:
Surface roughness is measured to explain problems of the other techniques. Is there a more quantitative measure to allow us use the surface roughness in relation to those techniques?
What is the use of surface roughness if it is performed after all other analyses, and especially TOF-SIMS that enhances roughness (usually produces waving surfaces, despite its low damage threshold)? Even more, cluster-Ar cleaning used for 3D profiling of Area I where all the other techniques have been performed, further enhances this damage.
Also, what is the laser beam properties of the instrument measuring roughness or the integration time during surface mapping and how it compares (i.e., with reference to sample damage) with the micro-Raman power densities? Is there a quantitative figure of merit coming out from these measurements that would result to better analytical practices?
Please, clarify the above or elaborate more on why this order of analysis is selected.
Lines 166-173:
A graphical map of the chronological order of all analyses would be useful for the reader to follow the manuscript. When the order is described in the paper, then this would be useful as reference methodology for future studies. Also, it would better clarify the previous question.
Lines 284-286:
The mass resolution of 5000 is given in general terms but then precise values are given, so we suggest that these two sentences are connected, removing the reference to the general value of 5000 which anyway does not add much information.
Lines 286-287:
Which is the kind of calibration curves? Do you possibly mean “mass calibration”? Please, clarify. Is there any reference paper to the procedure of creating these “calibration curves”?
Lines 296-307:
We understand that this work comes from reference [58]. This reference is not easily accessible, and because the procedure described in these lines is not clear in this manuscript, it is advised that this part is better elaborated. It is also not clear if the molecular ions on line 300 are considered to be contamination. The sentence in brackets on lines 305-307 should be integrated to the text, i.e. remove the brackets.
Line 393:
Describe the experimental constrain. The previous sentence does not indicate one. Probably you mean the diffusion due to roughness, explained afterwards. Please, rephrase these sentences in order to explain the problem after mentioning it. “This” in the sentence requires that you already mentioned the problem, not afterwards, else it is confusing.
Figs 5 and 6:
Also, looking at the elaboration in the text of the manuscript. Quality of spectra is poor due to roughness--as mentioned earlier. It would be nice to include with the spectra quality reference spectra of the same phases as well. The manuscript would be then self-contained and clearer. Backgrounds are bad and therefore further elaboration with also clearly indicating the reflections on all the graphs would assist reader also to make a reference to this work. Bands mentioned to the text should be marked with dotted lines on the graphs, as it is done for some cases.
Lines 445-447:
Lines do no read well because often the evidence is given after the claim and this is confusing.
Chapter 3.3:
Mid-IR reflectance spectra requires slightly better elaboration since it is not always easy to match the text with the graphs and the information is not effectively transferred and in cases looks vague. Please clarify results from here, interpreting that phases are amorphous, with results from Raman that show crystalline materials. How do these compare to each other?
Figure 7:
It would be of great assistance if the different mineralogical components are marked on the maps themselves as well without requiring referring to the caption (like it is in Fig. 9). Can you elaborate on the chemical composition of the chondrule?
Lines 814-816:
There are some reservations on the interpretation of phthalocyanine. This is a known pigment (blue colour). How is this interpreted to exist in a meteorite (or left to assume so)? Could it be contamination? We understand that there was a cleaning process by sputtering, but how secure is the above interpretation? Could it be contamination strongly bonded on the surface? This is only a single spot not found elsewhere and could very well be contamination. Could it be another interference? Quick combinatorial calculations have shown for example a couple pf Bi-based ions (Bi from the primary beam) such as 12C12-1H13-209Bi2, or a polar H17-C9-COO-Bi2. Although an explanation is given in Appendix B, it is suggested that further considerations are made. Since precise masses are missing, could these additional peaks are isotopomers of only Cu (both isotopes with high abundance), or with the addition of hydrogen.
Line 1023:
How do you conclude on iron sulphides when Raman shows Fe-hydro/oxides? Fig.9 indicates that where Fe is rich S is not that rich. Please, elaborate.
Line 1035:
How aliphatic and aromatic moieties are detected? Why is it so interpreted?
Lines 1067-1073:
Is the connection between Fe and organics not a bit speculative? Do you truly see a direct connection of these two or only because they coexist spatially? TOF-SIMS could create any kind of cluster and one requires to contact systematic studies to conclude in either way. Please, elaborate.
Author Response
Response to Reviewer 2 Comments
In black are the reviewer’s comments and questions. Our responses are in red. The highlighted sentences indicate the modifications added to the manuscript. All the line numbers, figure numbers, table numbers etc. we refer to in these answers correspond to the 1st version of the manuscript. Of course these numbers may change slightly in the revised version due to the modifications added.
This manuscript is a detailed technical description of the analytical and interpretation work performed in characterising a carbonaceous chondrite. This is of high value in establishing good practices in interpretation and very informative in making direct comparisons between techniques for the same purpose, especially when combining remote investigations with microscopic, detailed in situ analyses. This is interesting when the composition of the smaller bodies of our solar system is compared with meteorites, and this work adds to past analyses with other techniques, differentiating in some respect with them on characterising a totally untreated sample of the meteorite.
The manuscript uses very good English and no typos have been detected. In places, the text is too descriptive and might confuse the reader, but it is the technical aspect of the paper that causes this effect. It is not typical bold characters to be used for whole sentences, so this feature might have to be removed. Also, smaller fonts in captions should be made equal to the rest of the text of the caption. Finally, sentences in brackets should be removed and be treated as main text. Title and abstract reflect the text.
We removed the bold characters and brackets for some sentences and corrected the font in some captions.
This paper is suggested to be published with minor modifications and prior to some clarifications. Some points to be considered are the following:
Paragraph 1 (Introduction):
In the introduction there is a list of references covering major techniques employed in characterising the organic matter of carbonaceous chondrites. The authors could also consider together with reference [19] to mention the following reference which is along the same lines, however describing a different setup with a special use of a new, segmented ion trap:
· Sabbah et al, 2017, Identification of PAH Isomeric Structure in Cosmic Dust Analogues: the AROMA setup, Astrophys J. 2017 Jul 1; 843(1): 34. doi: 10.3847/1538-4357/aa73dd
Also, in lines 152-155, where references are given to the Stardust mission, of interest might be the reference doing the same analysis with TOF-SIMS but rather using C60 as an ion beam, optimum for heavy organics compounds. Please, look at the following:
· Rost et al, 2008, C60 ToF-SIMS: A tool for high-resolution mapping of elements and organic compounds. Geochimica et Cosmochimica Acta 72(12).
We added both references suggested by the reviewer.
Lines 180-187:
Surface roughness is measured to explain problems of the other techniques. Is there a more quantitative measure to allow us use the surface roughness in relation to those techniques?
What is the use of surface roughness if it is performed after all other analyses, and especially TOF-SIMS that enhances roughness (usually produces waving surfaces, despite its low damage threshold)? Even more, cluster-Ar cleaning used for 3D profiling of Area I where all the other techniques have been performed, further enhances this damage.
Also, what is the laser beam properties of the instrument measuring roughness or the integration time during surface mapping and how it compares (i.e., with reference to sample damage) with the micro-Raman power densities? Is there a quantitative figure of merit coming out from these measurements that would result to better analytical practices?
Please, clarify the above or elaborate more on why this order of analysis is selected.
The surface roughness measurement reported in this manuscript was not aimed to assess the effect of the SIMS sputtering on the sample surface roughness. Given the doses and irradiation time applied, this effect happens at a spatial scale (the nanometric scale) that is much lower than the typical roughness of this sample (1 - 2 µm) and would not be efficiently assessed with this sample.
The main reason for this roughness measurement was to be sure that our TOF-SIMS analyses and mappings were sound and not biased by some shadowing effects that occur when surfaces with strong roughness are measured with the same set up, as explained in appendix A (lines 1205-1215). We added the reference to the appendix explanation in section 2.1 to be more clear about our intentions with this measurement.
The laser wavelength of the Keyence microscope used is 408 nm and the manufacturer indicates an emission power of 0.95 mW. We added these details to section 2.1. Contrary to the Raman instrument, the power emitted by the Keyence laser is fixed, it cannot be lowered, and as the spatial resolution of the microscope goes down to the nm, it could imply quite high power densities on the sample that could easily damage or alter its organic content especially. Without further control on the laser and further testing on some organic standards, it would have been too risky to perform the roughness measurement on the meteorite sample before the sequence of analyses were performed.
Lines 166-173:
A graphical map of the chronological order of all analyses would be useful for the reader to follow the manuscript. When the order is described in the paper, then this would be useful as reference methodology for future studies. Also, it would better clarify the previous question.
At the end of the paragraph, we added a sentence stating the chronological order of the analyses applied.
Lines 284-286:
The mass resolution of 5000 is given in general terms but then precise values are given, so we suggest that these two sentences are connected, removing the reference to the general value of 5000 which anyway does not add much information.
We removed the unnecessary general value.
Lines 286-287:
Which is the kind of calibration curves? Do you possibly mean “mass calibration”? Please, clarify. Is there any reference paper to the procedure of creating these “calibration curves”?
Yes, mass calibration is intended here. The procedure is similar to the one applied in Noun et al. 2016 and Brunelle et al. 2005. We clarified the sentence and added the references at these lines.
Lines 296-307:
We understand that this work comes from reference [58]. This reference is not easily accessible, and because the procedure described in these lines is not clear in this manuscript, it is advised that this part is better elaborated. It is also not clear if the molecular ions on line 300 are considered to be contamination. The sentence in brackets on lines 305-307 should be integrated to the text, i.e. remove the brackets.
We modified and clarified this paragraph by adding a few information on the way the cleaning procedure was monitored. Basically, measurements on the chondrules were used to monitor and optimize the sputtering time necessary to eliminate the organic surface contaminants. When the superficial deposition of organic contaminants was eliminated by the cleaning process (typically the related ions disappeared after a few 10s of irradiation), we observed simultaneously an increase then a plateau in ion emission rates of elemental ions such as Mg+, Si+, Fe+ in positive mode, and O-, S- and lowly complex compounds such as SiO2-, PO2- in negative mode. Besides, the organic ions mentioned at line 300 are not due to contamination. They do not disappear; they reach a steady state of intensity for the following analysis which lasted 900s of sputtering.
Line 393:
Describe the experimental constrain. The previous sentence does not indicate one. Probably you mean the diffusion due to roughness, explained afterwards. Please, rephrase these sentences in order to explain the problem after mentioning it. “This” in the sentence requires that you already mentioned the problem, not afterwards, else it is confusing.
The technical or experimental constrain we are referring to in this sentence, is just the surface roughness of the sample mentioned line 391. The poor baselines that can be noticed in Figure 5 are due to this surface roughness and complicate the spectra interpretation in the 2 – 9 µm range especially, because it can be difficult to distinguish real small IR bands from the bad baseline behavior in that IR range.
The sentence is rephrased this way “Despite the surface roughness that induces poor and noisy baselines, it was still possible to measure IR spectra that can be interpreted.
Figs 5 and 6:
Also, looking at the elaboration in the text of the manuscript. Quality of spectra is poor due to roughness--as mentioned earlier. It would be nice to include with the spectra quality reference spectra of the same phases as well. The manuscript would be then self-contained and clearer. Backgrounds are bad and therefore further elaboration with also clearly indicating the reflections on all the graphs would assist reader also to make a reference to this work. Bands mentioned to the text should be marked with dotted lines on the graphs, as it is done for some cases.
Infrared band position ranges of the C-O carbonate stretching band, the S-O sulfate stretching band, the Si-O silica stretching band, the Si-O silicate stretching bands, and O-H stretching and bending modes are well known and easily identified by a specialist measuring a chondrite sample. Databases and papers (such as J. W. Salisbury’s ones) do exist with measurements of pure standards. In this manuscript, because of the spatial resolution of the IR measurement and the fact that it is a reflectance measurement made on a rough surface, we cannot identify precise mineralogical compositions without further evidences from the other instruments. For example, in these measurements, it is easy to spot the sulfate and the carbonate signature, but it is impossible without the help of the other instruments to be sure that it is Calcium rich sulfates and carbonates. The spectra showed in Figure 5 and 6 are there to show the typical kind of IR signatures obtained. These spectra correspond to the typical signature of a chunk of the Paris meteorite when measured by IR reflectance. In the future, those spectra could be compared to those of other chondrite samples or asteroidal samples from sample return missions that would be measured in the same conditions, i.e. without any surface preparation.
We added the dotted lines on some graphs.
Lines 445-447:
Lines do no read well because often the evidence is given after the claim and this is confusing.
We changed the order of the sentences for a better understanding.
Chapter 3.3:
Mid-IR reflectance spectra requires slightly better elaboration since it is not always easy to match the text with the graphs and the information is not effectively transferred and in cases looks vague. Please clarify results from here, interpreting that phases are amorphous, with results from Raman that show crystalline materials. How do these compare to each other?
The Mid-IR spectra of the matrix are dominated by the a, b,c and d signatures showed in Figure 5. Spectra of a and b types are due to sulfate and carbonate, the strong bands showed in each spectrum are typical of respectively the S-O stretching band of sulfates and the C-O stretching bands of carbonates. Spectra of c and d types are due to a silicate component largely wide spread in the matrix. The absence of a multiband structure (except for the “companion” band at 11.3 µm), the position, and the very broad shape of the principal Si-O stretching band indicates that it belongs to a disordered silicate structure, ie an amorphous silicate, as also found by Dionnet et al. (2018). We added this explanation of the amorphous silicate attribution to the text. Further justification for the amorphous silicate attribution is then discussed in section 4.2, lines 922-932.
Those a, b, c and d dominant signatures reported in the Mid-IR section do not exclude the presence of some minor components, including the presence of crystalline silicate phases in the matrix. As an example of these latter, we show the signature of a forsterite measured by IR and found in the matrix, and the “d” mapping in Figure 7 show that some Mg-rich pyroxene is present in some localized areas in the matrix. Raman measurements on the other hand, are not able to assess the amorphous silicate presence, nor do they inform on the presence of hydration bands. The technique is biased towards the detection of crystalline silicate signatures. Besides, the forsterite and enstatite identified in the spectra shown in Figure 8, correspond to points 10 and 17 which both belong to chondrules. For the matrix, calcite, Ca-sulfate, iron hydroxide+oxide and the D and G bands of a polyaromatic carbon component are the only species identified by the few Raman measurements performed (Contrary to IR, we did not perform a full mapping of area I). We modified the sentence at line 512-515 to be clearer about the Raman outcome.
Figure 7:
It would be of great assistance if the different mineralogical components are marked on the maps themselves as well without requiring referring to the caption (like it is in Fig. 9). Can you elaborate on the chemical composition of the chondrule?
We added the panels’ titles as suggested by the reviewer for Figure 7.
As stated line 455, only one chondrule has been explored by the IR synchrotron measurements and only partially. The synchrotron measurement is the one with the best spectral quality, but still it is not good enough to have a full comprehension of the chondrule mineralogical composition, except for the few information given at lines 455-461. Besides, as the main focus of the paper was rather the matrix because of its organic content, we did not try to improve our IR measurements and make specific and adapted mappings on the chondrules to better decipher their compositions.
Lines 814-816:
There are some reservations on the interpretation of phthalocyanine. This is a known pigment (blue colour). How is this interpreted to exist in a meteorite (or left to assume so)? Could it be contamination? We understand that there was a cleaning process by sputtering, but how secure is the above interpretation? Could it be contamination strongly bonded on the surface? This is only a single spot not found elsewhere and could very well be contamination. Could it be another interference? Quick combinatorial calculations have shown for example a couple pf Bi-based ions (Bi from the primary beam) such as 12C12-1H13-209Bi2, or a polar H17-C9-COO-Bi2. Although an explanation is given in Appendix B, it is suggested that further considerations are made. Since precise masses are missing, could these additional peaks are isotopomers of only Cu (both isotopes with high abundance), or with the addition of hydrogen.
First, we are not attributing the [C32H16N8Cu]+ compound to phthalocyanine. We only gave the phtalocyanine name to help the reader at imagining a possible structure for the [C32H16N8Cu]+ compound (Lines 816-819).
Second, the phtalocyanine pigment is not a common contaminant that one would easily and randomly find when performing a TOF-SIMS measurement. Great care was of course taken to handle the analyzed sample before, during and after the TOF-SIMS analyses. There was no contact between the sample surface and any other surface. Prior to each TOF-SIMS analysis, the sample surface was cleaned from any superficial contamination by the argon cluster beam as explained in section 2.6 and some TOF-SIMS control analyzes were performed during the study without finding any new contaminations or significant modification of the detection.
Third, copper is a very minor element present in chondrites’ matrixes. Finding larger regions or multiple spots containing the [C32H16N8Cu]+ would have been very suspect and would have pointed toward a contamination problem.
Fourth, the formulas suggested with two Bi atoms are very unlikely because the corresponding adducts with one Bi are not present in the same spectrum. However, it is quite possible to have a contribution from C32H17N8Cu+, i.e. the ion containing one more hydrogen, to the peaks at m/z 576.08, 577.08 and 578.08 u. We added this suggestion in appendix B, at lines 1224-1225.
Line 1023:
How do you conclude on iron sulphides when Raman shows Fe-hydro/oxides? Fig.9 indicates that where Fe is rich S is not that rich. Please, elaborate.
Indeed, the regions discussed line 1023 are particularly rich in Fe. They are not “hot spots” for S like the sulfate-containing regions are (Figure 9), however they still contain the average matrix concentration in S of ~45000 µg/g (at line 975, the S/Si ratio of 0.35 is given). Sulfur in those particular regions do not belong to sulfates according to IR, Raman and TOF-SIMS measurement. However, it must be bonded to something, and iron seems to be likely if we consider Figure 13 and the way HS- and Fe+ mappings fit very well. This latter assumption does not imply that all Fe+ in the discussed region are bonded to sulfur. On the contrary, lines 971-981 explain how the high Fe+ concentration in those regions must be due to different contributions: sulfides (very likely**), goethite with a possible contribution from hematite (indubitably, due to IR and Raman identifications), metallic iron (possible and expected but the techniques applied here cannot identify it as a mineral phase) and silicates containing iron (possible and expected but the techniques applied here are not suited to decipher the mineralogical composition of the detected silicates). We forgot to mention this latter possibility at line 1023. Thus, we corrected the sentence that states now “According to PIXE and TOF-SIMS measurements, these specific regions most likely contain metallic iron, and/or silicates containing iron, and/or iron sulfide, and according to IR and Raman results, part of it has probably been altered by water to form goethite and possibly hematite.”
** Raman measurements were scarce and correspond to 1µm-size spots each. It is not possible to conclude here that there would be no sulfides in the matrix based on their absence in the very few spectra measured.
Line 1035:
How aliphatic and aromatic moieties are detected? Why is it so interpreted?
In TOF-SIMS results, the detection of specific highly unsaturated organic ions in positive and negative mode, their comparison to PAH standard measurements, and their attribution to aromatic moieties is presented and discussed in section 3.6.2, at lines 761-771. The corresponding mappings are in Figure 15 (panels: Sum of C2nH2n-1+, Sum of C2nH2n-1+, Sum of CxH- and Sum of Cx-). We added the reference to the figure at line 761. Besides, in section 3.4, Raman results clearly indicate the presence of disordered polyaromatic carbon spread throughout the matrix and easily identified by their typical D and G bands (lines 513-519).
As for the aliphatic moieties, a large part of the organic ions detected by TOF-SIMS and presented throughout section 3.6.2 (such as C2H5+, CH4N+, C2H6N+ and many others) are either almost saturated, or containing too much heteroatoms, or both, and cannot belong to aromatic moieties. Thus, at line 1033-1034, we carefully state that “the ion fragments detected by TOF-SIMS are consistent with structures composed of a “nonpolar” carbon skeleton or “backbone” randomly branched and containing aliphatic and aromatic moieties; and polar functions distributed all over the carbon skeleton.”.
Lines 1067-1073:
Is the connection between Fe and organics not a bit speculative? Do you truly see a direct connection of these two or only because they coexist spatially? TOF-SIMS could create any kind of cluster and one requires to contact systematic studies to conclude in either way. Please, elaborate.
In section 3.6.2, at lines 797-800, it is clearly mentioned that “The detection of these ions, and especially the heaviest ones, is not only another proof of the strong spatial correlation between the organic content and iron as stated at the beginning of this section, but it may also indicate that there are strong bonds between iron and organic moieties in the meteorite matrix”. Contrary to Ag or Au that are metals known to easily form cluster ions and adducts when subjected to primary ions bombardment, iron adducts are not easy to form, especially for fragments as large as the ones we report. They need a molecular proximity between iron atoms and the organic fragments, and a certain homogeneity of both components in the considered zone. Whether the molecular proximity implies covalent bonds or other binding mechanisms such as the ones described lines 1044-1050, do not change the fact that the connection between those iron atoms and those organic moieties exists and is strong enough in those specific regions to allow the detection of Fe-containing large organic moieties.
